# Active Confusion Expression in Large Language Models: Leveraging World Models toward Better Social Reasoning

## Abstract

While large language models (LLMs) excel in mathematical and code reasoning, we observe they struggle with social reasoning tasks, exhibiting cognitive confusion, logical inconsistencies, and conflation between objective world states and subjective belief states. Through detailed analysis of DeepSeek-R1's reasoning trajectories, we find that LLMs frequently encounter reasoning impasses and tend to output contradictory terms like "tricky" and "confused" when processing scenarios with multiple participants and timelines, leading to erroneous reasoning or infinite loops. The core issue is their inability to disentangle objective reality from agents' subjective beliefs. To address this, we propose an adaptive world model-enhanced reasoning mechanism that constructs a dynamic textual world model to track entity states and temporal sequences. It dynamically monitors reasoning trajectories for confusion indicators and promptly intervenes by providing clear world state descriptions, helping models navigate through cognitive dilemmas. The mechanism mimics how humans use implicit world models to distinguish between external events and internal beliefs. Evaluations on three social benchmarks demonstrate significant improvements in accuracy (e.g., +10% in Hi-ToM) while reducing computational costs (up to 33.8% token reduction), offering a simple yet effective solution for deploying LLMs in social contexts.

## 1 Introduction

With the rapid development of large language models (LLMs), their reasoning capabilities have achieved significant improvements, particularly in reasoning domains such as mathematics and code generation. Notable examples include OpenAI's o1 (Jaech et al., 2024), DeepSeek-R1 (Guo et al., 2025), Qwen-QWQ (Team, 2024; 2025), and Claude Sonnet 4 (Anthropic, 2025), which demonstrate substantial knowledge and logical reasoning abilities through extended chain-of-thought (CoT) (Wei et al., 2022) processes.

However, when confronted with social reasoning tasks, LLMs exhibit significant limitations, e.g., **cognitive confusion** when processing multiple timelines, **logical inconsistencies** when analyzing complex character relationships, and **conflation** between **objective world states** and **subjective belief states** in social scenarios with multiple participants. These challenges significantly hinder the deployment of LLMs in social contexts.

Unlike mathematical reasoning, which requires extensive domain knowledge and mathematical logic, social reasoning necessitates that models comprehend real-world events occurring at different temporal points, disambiguate relationships among multiple involved agents, and establish connections between agents' subjective beliefs and objective world states. While current reasoning LLMs have demonstrated substantial improvements on mathematical reasoning, their social reasoning behavior often conflates participants' subjective beliefs with objective reality, generating verbose, meaningless, and logically inconsistent CoT, resulting in poor efficiency and low accuracy.

Specifically, objective world states represent real-world events that occur physically, e.g., one object being moved or some agents leaving the room, while subjective belief states represent characters mental thoughts about other participants (agent or objects). Since each agent can only observe partial information, their beliefs about unobserved event may become misaligned with reality. A

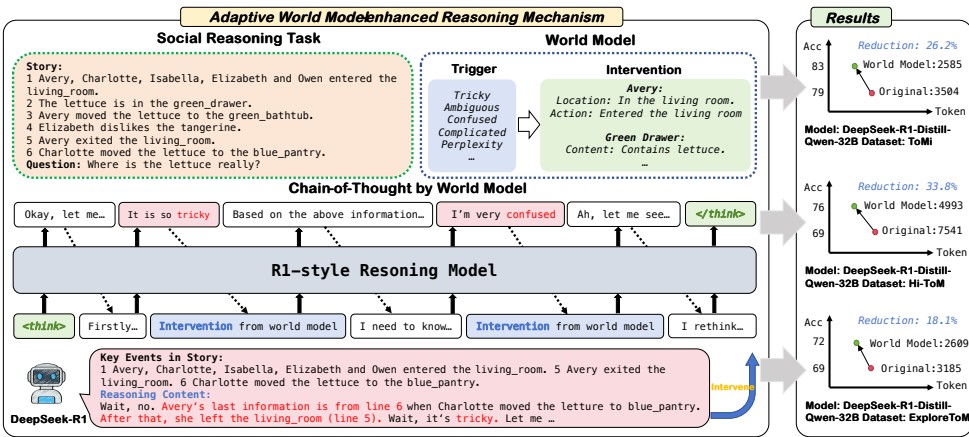

Figure 1: Our adaptive world model-enhanced reasoning mechanism. The system detects confusing words and adaptively use world model to intervene the reasoning trajectories, achieving significant improvements in both accuracy and token efficiency across three social reasoning benchmarks.

canonical example of is demonstrated in Theory of Mind (Premack & Woodruff, 1978): *After an agent leaves the room, they remain unaware of subsequent changes that occur within the room (e.g., the apple being moved from the refrigerator to the basket), thus their subjective beliefs about objects in the room will diverge from objective reality, i.e., they persistently believe the apple remains in the refrigerator.* We find this conflation between reality and beliefs constitutes the primary cause of reasoning failures, particularly in scenarios involving multiple participants and multiple timelines. However, such scenarios are ubiquitous in everyday life.

In this study, we first conduct a detailed analysis of DeepSeek-R1's reasoning trajectories in social reasoning tasks, investigating the underlying causes of its confusion in their reasoning thoughts. We observe that in the initial stages of reasoning, the model typically can analyze story contexts and clarify character relationships, demonstrating normal cognitive capabilities. However, when the reasoning process involves multiple participants (both agents and objects) and events occurring at different temporal points, the model is prone to falling into cognitive dilemmas, manifesting as confusion and disorientation. Notably, as shown in Figure 1, we find LLMs tend to output **contradictory terms** such as "tricky", "ambiguous" and "confused" under these circumstances, ultimately leading to erroneous reasoning results or infinite thinking loops.

Through careful analysis of reasoning thoughts at these "confusion" moments, we find the LLMs have fallen into cognitive dilemmas, conflating objective states in the real world with agents' subjective beliefs, and struggling to disentangle the relationships between them. To address this, we revisit human social cognitive processes. When humans navigate daily interactions, they naturally construct an **implicit world model** to track entity states, character relationships, and temporal sequences of each event. This cognitive model enables humans to easily distinguish and disentangle the relationships between external events and each agent's internal beliefs, thereby facilitating better understanding of others' intentions, goals, and emotions for social interaction.

Inspired by this, we propose an adaptive world model-enhanced reasoning mechanism, augmenting social reasoning through the simultaneous construction of a textual world model. As shown in Figure 1, our mechanism establishes a dynamic world model that tracks entities and task states in social events. It continuously monitors LLMs' reasoning trajectories, providing timely interventions when the model encounters reasoning confusion or cognitive biases. It provides clear world state descriptions to help the model navigate through confusion and break free from reasoning impasses. Our mechanism consists of two components:

- **Trigger mechanism**: The system monitors contradictory words such as "tricky", "ambiguous" and "confused" in their reasoning trajectory (blue square in Figure 1). When such scenarios are detected, the world state intervention is activated.

- **Intervention process**: Once intervention is triggered, the LLMs immediately halt its previous "confused" reasoning and instead retrieves world states (including entity states, char-

acter states, and timelines) from the latest world model. These states are then inserted after the contradictory terms, guiding LLMs to reflect on their previous reasoning dilemmas and return to correct trajectories.

Similar to how humans construct implicit world models in their brain, this self-constructed world model enables LLMs to re-examine their current thinking state, clarify relationships between characters and entities, and break out of existing reasoning dilemmas. This design effectively improves LLMs' social reasoning accuracy and consistency while reducing token consumption.

Our contributions are threefold: (1) We identified significant issues with LLMs in social reasoning tasks, including cognitive confusion, logical inconsistency, and frequent conflation between objective world states and subjective belief states. (2) We proposed an adaptive world model-enhanced reasoning mechanism that leverages self-constructed textual world models to enable active intervention, helping models overcome reasoning impasses. (3) We conducted comprehensive evaluations on three representative benchmarks ToMi (Le et al., 2019), Hi-ToM (Wu et al., 2023), and Explore-ToM (Sclar et al., 2024), validating the effectiveness of our method in **improving accuracy** and **reducing computational costs**.

## 2 RELATED WORK

**Strategies for Enhancing LLMs' Cognitive Development in Social Domain** To improve the performance of LLMs in social reasoning tasks, existing methods can be broadly categorized into three main directions: (1) Prompt-based Methods, (2) Tool-based Methods, and (3) Model-based Methods. In the field of prompt engineering, SimToM (Wilf et al., 2024) enhances social reasoning capabilities by designing specific prompting strategies that guide models to perform perspective-taking. PercepToM (Jung et al., 2024) focuses on optimizing the conversion process from perceptual information to belief inference through refined contextual information extraction. Question-Analysis Prompting (QAP) (Yugeswardeenoo et al., 2024) guides models to first conduct an in-depth analysis and understanding of the problem. The "problem analysis first" strategy helps models better grasp the key information. Meanwhile, Huang et al. (2024) employs LLMs themselves as world model state trackers to monitor changes in environmental entity positions and character belief processes. Hou et al. (2024) designed specialized belief-solving mechanisms that decompose complex higher-order reasoning tasks into simpler lower-order cognitive problems through set operations in the temporal dimension. MemoRAG (Qian et al., 2025) uses a Memory LLM model to generate answer clues, which then guide retrieval tools to locate relevant passages, demonstrating excellent performance in handling long contexts and complex tasks. As for model-based methods, SymbolicToM (Sclar et al., 2023) introduces symbolic graphical representations to model and track the dynamic changes in character beliefs. AutoToM, MMToM, and MuMA-ToM (Zhang et al., 2025; Shi et al., 2025; Jin et al., 2024) adopt Bayesian inference frameworks, addressing uncertainty issues in social reasoning from a probabilistic modeling perspective. Although existing research has made significant progress in the field of social reasoning, deficiencies remain in dynamic intervention during reasoning. Our method addresses the challenges in social reasoning through cognitive intervention strategies.

**Test-time scaling** Test-time scaling can be primarily categorized into three aspects: (1) Reward-guided efficient reasoning, (2) Confidence and certainty-based adaptive reasoning, and (3) Consistency-based selective reasoning. Speculative Rejection (Sun et al., 2024) optimizes the computational overhead of Best-of-N decoding by evaluating the quality of partially generated results using reward models. Reward-Guided Speculative Decoding (Liao et al., 2025) improves speculative decoding methods by selectively accepting high-quality results using Process Reward Models (PRM). Meanwhile, Dynamic Parallel Tree Search (Ding et al., 2025) addresses the efficiency issues of tree-based reasoning through parallelization optimization and search-transition mechanisms. FastMCTS (Li et al., 2025) employs Monte Carlo Tree Search to prioritize high-confidence trajectories for deep reasoning, improving multi-step reasoning data synthesis. Length-filtered Vote (Wu et al., 2025) utilizes length-aware majority voting methods, grouping answers according to CoT length. Self-Truncation Best-of-N (Wang et al., 2025b) enhances BoN sampling efficiency by introducing early termination mechanisms, using consistency to measure importance. However, existing research methods focus on post-hoc filtering or early termination of reasoning paths, lacking the abil-

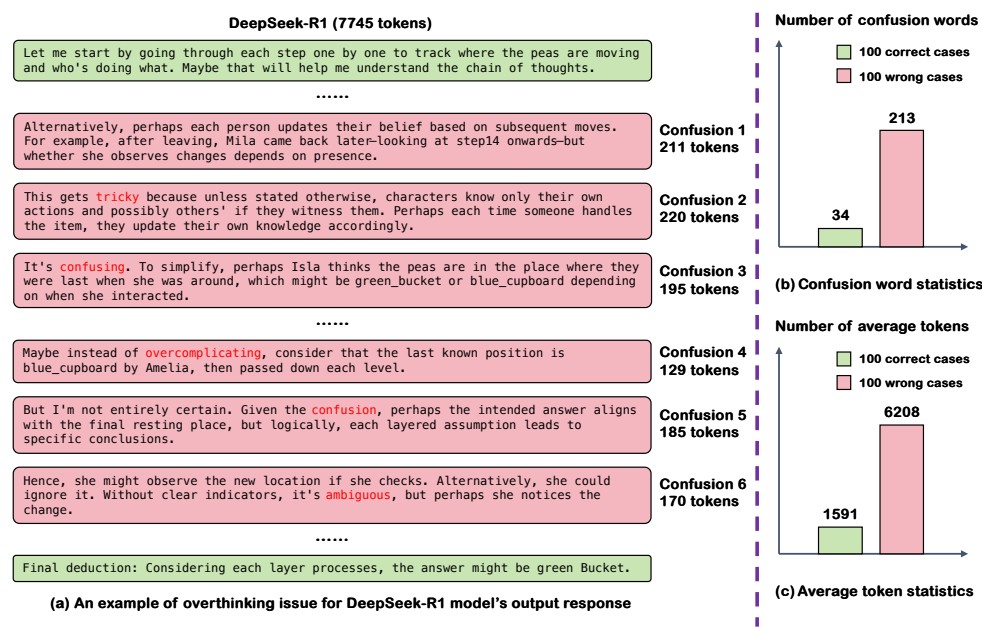

Figure 2: Analysis of DeepSeek-R1's reasoning dilemmas in social reasoning tasks. (a): Overthinking examples with confusion words. (b) and (c): Statistics of confusion words and average token consumption from 100 correct cases and 100 wrong cases.

ity to proactively inject structured guidance to help models escape cognitive dilemmas. Our method addresses reasoning inefficiency at its source by providing structured contextual intervention.

## 3 METHOD

### 3.1 PRELIMINARY

#### 3.1.1 EXPLORATION OF DEEPSEEK-R1'S REASONING TRAJECTORIES

We first collected DeepSeek-R1's reasoning trajectories on social reasoning datasets, focusing on the model's output characteristics when processing complex reasoning tasks. Subsequently, we conducted observational analysis from two perspectives: (1) qualitative analysis of linguistic expressions in the model's output; (2) quantitative statistics of confusion word counts and average token consumption in correct and wrong cases.

As demonstrated in the case shown in Figure 2(a), we observed a significant phenomenon: when DeepSeek-R1 encounters complex social reasoning scenarios, its output begins to frequently exhibit numerous confusion words such as "tricky" and "confusion". Furthermore, it will be trapped in a cycle of repeatedly questioning its own reasoning results, with descriptions such as "Alternatively, perhaps..." or "Maybe instead of...", indicating that the model cannot make clear judgments between objective world states and subjective belief states.

Building on this foundation, we discovered that the more pronounced these phenomena become in DeepSeek-R1, the higher the probability of incorrect answers. Therefore, we selected 100 correct cases and 100 wrong cases to analyze the quantity of confusion words and average token consumption, as shown in Figure 2(b) and (c). The wrong answer cases exhibited significantly higher numbers of confusion words and average token consumption compared to the correct answer cases. This indicates that when facing complex and error-prone problems, the model is more likely to fall into cognitive dilemmas, and such dilemmas require the introduction of targeted interventions to resolve.

#### 3.1.2 ANALYSIS OF CONFUSION WORDS

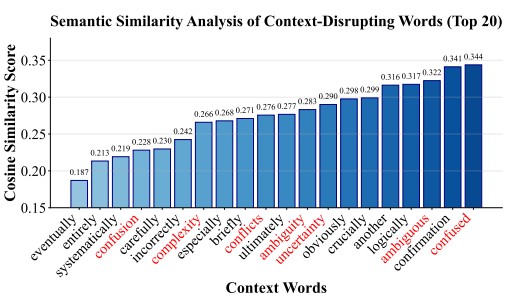

Figure 3: Semantic similarity distribution of candidate interruption words in DeepSeek-R1's reasoning trajectories.

Based on the above exploration results, we found that the number of confusion words is positively correlated with the accuracy of reasoning results, so we further investigate the impact that the appearance of confusion words has on the reasoning process. We conducted contextual semantic analysis on each word in the DeepSeek-R1 reasoning trajectory, selecting cosine similarity between preceding and following context as the metric. Then we removed words that obviously do not conform to semantic transitions, and finally selected the top 20 words with the lowest similarity from the remaining words. As shown in Figure 3, we observed many confusion words that frequently appeared in Section 3.1.1, which indicates that confusion words not only represent confusion themselves but also increase the semantic differences between the preceding and following context, thus leading to higher error rates.

## 3.2 ADAPTIVE WORLD MODEL-ENHANCED REASONING WITH COGNITIVE INTERVENTION

Given the issues identified in DeepSeek-R1's reasoning trajectories as discussed in the previous section, we propose a framework based on adaptive world model-enhanced reasoning with cognitive intervention to address the challenges in social reasoning tasks. First, by employing intervention words to trigger proactive cognitive intervention, timely interrupting the model when it encounters reasoning dilemmas. Second, introducing an adaptive world model that provides character and entity state information to guide the model back to coherent reasoning trajectories. Specifi-

Table 1: Keywords Representing Confusion States

| **Intervention Words** |
| --- |
| "ambiguous", "complicating", "confusion", "confusing", "confused", "perplexity", "puzzle", "puzzled", "puzzling", "perplexed", "complication", "troubled", "tricky", "conflicts", "ambiguity" |

cally, our approach begins with intervention word selection based on perplexity analysis, followed by constructing a world model that maintains entity and character states. When intervention words appear during the reasoning process, the system adaptively intervenes and injects state information to help the model distinguish between objective world states and subjective belief states, ultimately achieving globally consistent reasoning capabilities.

### 3.2.1 COGNITIVE INTERVENTION

In Section 3.1.2 , we discovered that confusion words not only represent confusion themselves but also increase the semantic differences between the preceding and following text, ultimately affecting normal reasoning. Therefore, we believe that effective intervention is needed to break this state when reasoning is in similar situations. Wang et al. (2025a) conducted research on where to interrupt reasoning. Based on the above assumption, we compared the perplexity and semantic similarity between the interruption words they mentioned and the aforementioned confusion words. As shown in the Figure 4, confusion words have high perplexity while maintaining low semantic similarity, making them suitable as reasoning interruption points. We name these words as intervention words, as shown in the Table 1.

### 3.2.2 CONSTRUCTION OF WORLD MODELS

We retain the model's reasoning trajectory before the intervention words and provide it with the information from the world model to guide its thought process. The world model here refers to a set of structured textual world model that can explicitly describe the temporal causal relationships between objective world states and subjective belief states. It aims to address issues that arise in the social reasoning process, as well as to reduce token redundancy. And finally helps guide the model back to unconfused reasoning trajectories.

As illustrated in Figure 5, the world model maintains a temporal state, dynamically updating the entities and characters state by progressively processing behavioral events in narrative actions. When the model encounters confused thinking and logical inconsistency during the reasoning process, the system adaptively selects corresponding states from the constructed world model for intervention. The state adopts a structured format: "<information> {state from world model} </information>", which encapsulates the current states of entities and characters. When reasoning difficulties persist, the model will reselect the required corresponding states based on the current reasoning state. We introduce a parameter $k$ as the maximum number of interventions allowed by the world model. This world model aims to provide active leading for reasoning processes in confused states, guiding the model to accurately distinguish between objective world states and subjective belief states, thereby constructing a unified temporal logical framework and achieving globally consistent reasoning capabilities.

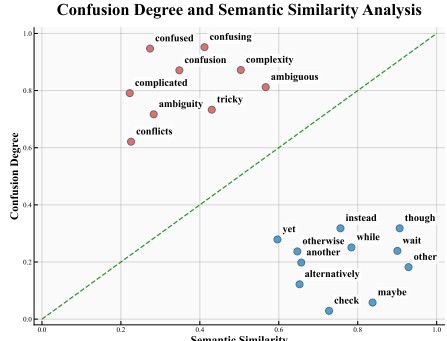

Figure 4: Comparison of confusion degree and semantic similarity between candidates and traditional interruption words.

After receiving information from world model, the model re-examines previous reasoning trajectory and conducts secondary reasoning, systematically analyzing narrative process, clarifying relationships between characters and entities, thereby forming a clear thinking logic. This effectively corrects the confusion between objective world states and subjective belief states, ultimately returning to the correct reasoning path. Our method not only reduces the number of tokens consumption, but improves the accuracy of the social reasoning significantly.

## 4 EXPERIMENT

In this section, we evaluate our method across multiple social reasoning tasks, such as ToMi (Le et al., 2019), Hi-ToM (Wu et al., 2023) and Explore-ToM (Sclar et al., 2024).

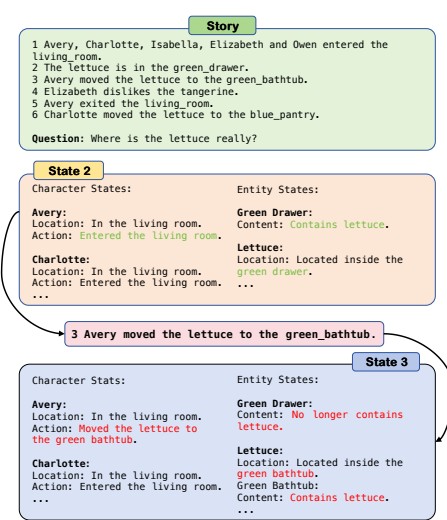

Figure 5: World Model Generation Process

### 4.1 IMPLEMENTATION DETAILS

We select three social reasoning datasets: ToMi, Hi-ToM, and ExploreToM. These three datasets encompass social reasoning tasks across multiple levels and demonstrate high representativeness.

Our experiment was implemented on Ubuntu 20.04.4 LTS system environment using the PyTorch framework. We selected multiple mainstream LLMs for performance comparison and evaluation, including DeepSeek-v3 (DeepSeek-AI, 2024), Llama-3.1 (Dubey et al., 2024) series, Qwen2.5 (Yang et al., 2024) series, GPT-5, o1-preview (OpenAI, 2024), DeepSeek-R1, Claude series, and Qwen3 (Qwen Team, 2024) series.

The experiment adopted accuracy and token consumption as primary evaluation metrics. The token consumption statistics include the tokens required for constructing the world model. Model configuration parameters were set as follows: maximum token length was set to 8192, temperature was set to 0.7, and repetition penalty was set to 1.2.

To thoroughly investigate the effectiveness of the adaptive world model-enhanced reasoning mechanism, we employed DeepSeek-R1-Distill-Qwen-32B (Guo et al., 2025), which possesses excellent reasoning capabilities and computational efficiency, as the world model generation model and core

Table 2: Performance of Models on Social Reasoning Datasets. For models with dual entries, the first row shows baseline performance while the second row shows results after incorporating adaptive world model-enhanced reasoning mechanism.

| Model | ToMi | | Hi-ToM | | ExploreToM | |
|---|---|---|---|---|---|---|
| | Acc | Token | Acc | Token | Acc | Token |
| *Non-reasoning LLMs* | | | | | | |
| DeepSeek-v3 | 76.00 | 726 | 69.68 | 2182 | 59.68 | 516 |
| Llama-3.1-8B-Instruct-Turbo | 64.68 | 442 | 58.00 | 1322 | 53.33 | 424 |
| Llama-3.1-70B-Instruct-Turbo | 72.00 | 408 | 66.33 | 1244 | 57.00 | 336 |
| Llama-3.1-405B-Instruct-Turbo | 77.33 | 356 | 68.00 | 1490 | 68.68 | 362 |
| Qwen2.5-7B-Instruct-Turbo | 60.00 | 392 | 45.67 | 1168 | 45.00 | 356 |
| Qwen2.5-32B-Instruct-Turbo | 73.67 | 422 | 60.33 | 952 | 54.67 | 470 |
| Qwen2.5-72B-Instruct-Turbo | 79.67 | 396 | 68.68 | 1548 | 55.00 | 402 |
| *Reasoning LLMs* | | | | | | |
| GPT-4o | 74.00 | 1270 | 70.00 | 2502 | 56.68 | 1412 |
| GPT-5 | 98.33 | 834 | 96.00 | 1768 | 93.67 | 1089 |
| o1-preview | 91.33 | 2866 | 93.68 | 6392 | 82.33 | 1730 |
| o3-mini | 73.00 | 1762 | 89.33 | 5666 | 74.00 | 2924 |
| DeepSeek-R1 | 93.33 | 2846 | 77.33 | 7710 | 78.00 | 2124 |
| Claude-3.5-sonnet | 88.00 | 349 | 83.67 | 571 | 75.00 | 416 |
| Claude-sonnet-4 | 97.33 | 597 | 96.67 | 892 | 81.67 | 668 |
| Qwen3-8B | 59.33 | - | 52.33 | - | 65.67 | - |
| Qwen3-14B | 60.67 | - | 54.00 | - | 68.00 | - |
| Qwen3-32B | 64.33 | - | 56.33 | - | 70.67 | - |
| DeepSeek-R1-Distill-Qwen-7B | 55.33 | 3552 | 43.33 | 6080 | 60.67 | 4087 |
| | 57.33$_{\uparrow 2.00}$ | 2669$_{\downarrow 883}$ | 49.33$_{\uparrow 6.00}$ | 4855$_{\downarrow 1225}$ | 61.67$_{\uparrow 1.00}$ | 3196$_{\downarrow 891}$ |
| DeepSeek-R1-Distill-Qwen-14B | 70.33 | 3706 | 59.00 | 7889 | 65.33 | 3097 |
| | 72.33$_{\uparrow 2.00}$ | 3147$_{\downarrow 559}$ | 61.33$_{\uparrow 2.33}$ | 6913$_{\downarrow 976}$ | 66.33$_{\uparrow 1.00}$ | 2286$_{\downarrow 811}$ |
| DeepSeek-R1-Distill-Qwen-32B | 79.33 | 3504 | 69.33 | 7541 | 69.67 | 3185 |
| | 83.67$_{\uparrow 4.34}$ | 2585$_{\downarrow 919}$ | 76.67$_{\uparrow 7.34}$ | 4993$_{\downarrow 2548}$ | 72.67$_{\uparrow 3.00}$ | 2609$_{\downarrow 576}$ |
| Qwen3-8B-think | 74.67 | 1713 | 74.67 | 3795 | 67.67 | 2359 |
| | 75.33$_{\uparrow 0.66}$ | 1446$_{\downarrow 267}$ | 74.67$_{\uparrow 0.00}$ | 3246$_{\downarrow 549}$ | 69.33$_{\uparrow 1.66}$ | 1994$_{\downarrow 365}$ |
| Qwen3-14B-think | 76.00 | 1619 | 83.33 | 4269 | 71.00 | 2061 |
| | 76.67$_{\uparrow 0.67}$ | 1336$_{\downarrow 283}$ | 85.00$_{\uparrow 1.67}$ | 3615$_{\downarrow 654}$ | 71.33$_{\uparrow 0.33}$ | 1551$_{\downarrow 510}$ |
| Qwen3-32B-think | 81.33 | 1942 | 88.00 | 3316 | 74.33 | 2193 |
| | 81.33$_{\uparrow 0.00}$ | 1887$_{\downarrow 55}$ | 88.67$_{\uparrow 0.67}$ | 2840$_{\downarrow 476}$ | 76.00$_{\uparrow 1.67}$ | 1805$_{\downarrow 388}$ |

experimental model to systematically explore the impact of different threshold $k$ parameters, alternative intervention words settings, and various methods on model performance.

## 4.2 MAIN RESULTS

We test the performance on both non-reasoning and reasoning models above in three social reasoning datasets. Moreover, we evaluated the performance of DeepSeek-R1-Distill series and Qwen3 series models with adaptive world model-enhanced reasoning mechanism, to comprehensively analyze the impact of our method on social reasoning capabilities.

**Reasoning LLMs significantly outperformed non-reasoning LLMs across all social reasoning datasets** As shown in Table 2, on the ToMi dataset, reasoning LLMs achieved a maximum accuracy of 98.33% (GPT-5), while the best performance of non-reasoning LLMs was only 79.67%, representing a performance gap of 18.66 percentage points. On the more challenging Hi-ToM dataset, this gap further expanded to 26.32 percentage points (96.00% vs 69.68%). These results demonstrate that reasoning mechanisms based on long chain-of-thought can significantly enhance model performance on complex social reasoning tasks.

**Adaptive world model-enhanced reasoning mechanism shows significant accuracy improvement** For the DeepSeek-R1-Distill series models, the 7B version improved from 55.33% to 57.33% (+2.00) on the ToMi dataset, the 14B version improved from 70.33% to 72.33% (+2.00), and the 32B version improved from 79.33% to 83.67% (+4.34). This indicates that the adaptive world model-enhanced reasoning mechanism can effectively enhance models' social reasoning capabilities, with more significant improvement effects observed for larger-scale models.

**Adaptive world model-enhanced reasoning mechanism reduces token consumption** DeepSeek-R1-Distill-Qwen-32B achieved token reductions of 26.2%, 33.8%, and 18.1% respectively. Similarly, the Qwen3-think LLMs also demonstrated varying degrees of token consumption reduction.

## 4.3 ANALYSIS OF MAXIMUM INTERVENTION COUNT

Table 3: Performance under different maximum intervention count $k$.

| Model | Threshold | ToMi | | Hi-ToM | | ExploreToM | |
|---|---|---|---|---|---|---|---|
| | | Acc | Token | Acc | Token | Acc | Token |
| DeepSeek-R1 Distill Qwen -32B | Baseline | 79.33 | 3504 | 69.33 | 7541 | 69.67 | 3185 |
| | $k=1$ | $80.00_{\uparrow 0.67}$ | $3196_{\downarrow 308}$ | $70.67_{\uparrow 1.34}$ | $6731_{\downarrow 810}$ | $70.33_{\uparrow 0.66}$ | $2971_{\downarrow 214}$ |
| | $k=2$ | $81.67_{\uparrow 2.34}$ | $2875_{\downarrow 629}$ | $73.33_{\uparrow 4.00}$ | $6119_{\downarrow 1422}$ | $71.00_{\uparrow 1.33}$ | $2554_{\downarrow 631}$ |
| | $k=3$ | $83.67_{\uparrow 4.34}$ | $2585_{\downarrow 919}$ | $76.67_{\uparrow 7.34}$ | $4993_{\downarrow 2548}$ | $72.67_{\uparrow 3.00}$ | $2609_{\downarrow 576}$ |
| | $k=4$ | $83.67_{\uparrow 4.34}$ | $2585_{\downarrow 919}$ | $77.67_{\uparrow 8.34}$ | $4149_{\downarrow 3392}$ | $73.33_{\uparrow 3.66}$ | $2360_{\downarrow 825}$ |
| | $k=5$ | $83.67_{\uparrow 4.34}$ | $2585_{\downarrow 919}$ | $79.33_{\uparrow 10.00}$ | $3421_{\downarrow 4120}$ | $73.33_{\uparrow 3.66}$ | $2360_{\downarrow 825}$ |

The experimental results in Table 3 reveal important characteristics of the adaptive world model-enhanced reasoning mechanism. As the maximum intervention count $k$ increases, the model demonstrates varying degrees of accuracy improvement across all datasets. This consistent improvement indicates that increasing the frequency of world model intervention can provide LLM with richer social cognitive information, thereby enhancing its reasoning capabilities.

**More complex social reasoning tasks require more world model interventions** For the relatively simple ToMi dataset, performance improvement peaked at $k$=3 (83.67%) and remained stable thereafter, exhibiting clear performance saturation. However, on the more challenging Hi-ToM dataset, performance continued to improve throughout the entire $k$ value range. This phenomenon indicates that complex social reasoning tasks can more effectively utilize frequent world model interventions.

**Appropriate interventions improves efficiency** On the three datasets, high $k$ value settings not only improved accuracy but also significantly reduced token consumption. This "dual optimization" effect demonstrates that the adaptive world model-enhanced reasoning mechanism actually enhances the model's reasoning efficiency by providing more guidance.

## 4.4 ANALYSIS OF INTERVENTION WORDS CATAGORIES

DeepSeek-R1-Distill-Qwen-32B achieved significant performance improvements with our selected intervention words. To compare the impact of intervention words selection, We compared several categories of commonly used intervention words described in Wan et al. (2025): Pause-Validation (referred to as PV) and Branch-Extension (referred to as BE) as comparisons.

- **Pause-Validation**: wait, check, make sure, hold on, verify, let me see, confirm, ensure, evaluate, examine.
- **Branch-Extension**: alternatively, another, instead, however, while, yet, though, rather, otherwise, on the other hand.

As shown in Table 4, our selected intervention words (Ours) achieve optimal performance in both accuracy and token efficiency across all datasets compared to other common intervention words (+PV and +BE). This indicates that compared to choosing simple pause-validation or branch-extension words for interruption, our method demonstrates good methodological stability and applicability, which also validates the conclusions in section 3.2.1.

Table 4: Performance of DeepSeek-R1-Distill-Qwen-32B with different intervention words. PV denotes pause-validation words, and BE indicates branch-extension words.

| Model | Method | ToMi | | Hi-ToM | | ExploreToM | |
|---|---|---|---|---|---|---|---|
| | | Acc | Token | Acc | Token | Acc | Token |
| DeepSeek-R1 Distill Qwen -32B | Baseline | 79.33 | 3504 | 69.33 | 7541 | 69.67 | 3185 |
| | +PV | 82.33$_{\uparrow 3.00}$ | 2830$_{\downarrow 674}$ | 72.67$_{\uparrow 3.34}$ | 6059$_{\downarrow 1482}$ | 70.00$_{\uparrow 0.33}$ | 2971$_{\downarrow 214}$ |
| | +BE | 81.67$_{\uparrow 2.34}$ | 2803$_{\downarrow 701}$ | 74.00$_{\uparrow 4.67}$ | 5750$_{\downarrow 1791}$ | 71.33$_{\uparrow 1.66}$ | 2781$_{\downarrow 404}$ |
| | Ours | 83.67$_{\uparrow 4.34}$ | 2585$_{\downarrow 919}$ | 76.67$_{\uparrow 7.34}$ | 4993$_{\downarrow 2548}$ | 72.67$_{\uparrow 3.00}$ | 2609$_{\downarrow 576}$ |

## 4.5 ANALYSIS OF OTHER METHODS

To validate the effectiveness of our method in improving reasoning performance, we compare our method with several approaches for enhancing model reasoning capabilities, including Chain-of-Thought (CoT), Tree of Thoughts (ToT)(Yao et al., 2023a), Reasoning and Acting (ReAct) (Yao et al., 2023b), and Reflexion (Shinn et al., 2023) for comparative evaluation.

Table 5: Comparison with different prompting strategy on social reasoning tasks.

| Model | Method | ToMi | | Hi-ToM | | ExploreToM | |
|---|---|---|---|---|---|---|---|
| | | Acc | Token | Acc | Token | Acc | Token |
| DeepSeek-R1 Distill Qwen -32B | Baseline | 79.33 | 3504 | 69.33 | 7541 | 69.67 | 3185 |
| | +CoT | 79.67$_{\uparrow 0.34}$ | 3456$_{\downarrow 48}$ | 70.00$_{\uparrow 0.67}$ | 7698$_{\uparrow 157}$ | 70.00$_{\uparrow 0.33}$ | 3156$_{\downarrow 29}$ |
| | +ToT | 80.33$_{\uparrow 1.00}$ | 3756$_{\uparrow 252}$ | 70.67$_{\uparrow 1.34}$ | 8123$_{\uparrow 582}$ | 70.67$_{\uparrow 1.00}$ | 3421$_{\uparrow 236}$ |
| | +ReAct | 81.00$_{\uparrow 1.67}$ | 2198$_{\downarrow 306}$ | 70.67$_{\uparrow 1.34}$ | 7234$_{\downarrow 307}$ | 69.67$_{\uparrow 0.00}$ | 3256$_{\uparrow 71}$ |
| | +Reflexion | 80.67$_{\uparrow 1.34}$ | 3076$_{\downarrow 428}$ | 71.33$_{\uparrow 2.00}$ | 6687$_{\downarrow 854}$ | 70.33$_{\uparrow 0.66}$ | 2934$_{\downarrow 251}$ |
| | Ours | 83.67$_{\uparrow 4.34}$ | 2585$_{\downarrow 919}$ | 76.67$_{\uparrow 7.34}$ | 4993$_{\downarrow 2548}$ | 72.67$_{\uparrow 3.00}$ | 2609$_{\downarrow 576}$ |

As shown in the Table 5, our method achieves significant performance improvements over other methods across all datasets. This demonstrates that our approach can more effectively handle the cognitive confusion in social reasoning tasks, and achieve dual improvements in both performance and efficiency.

## 5 CONCLUSIONS, LIMITATIONS AND FUTURE WORKS

In this paper, we identified and addressed critical limitations of current reasoning LLMs in social reasoning tasks. Through detailed analysis of DeepSeek-R1's reasoning trajectories, we discovered that models are prone to cognitive confusion, logical inconsistencies, and conflation between objective world states and subjective belief states when processing complex social scenarios. We reveals that LLMs frequently fall into cognitive dilemmas when encountering contradictory words such as "confused" and "ambiguous", leading to reasoning errors or multiple loops. To address this issue, we proposed an adaptive world model-enhanced reasoning mechanism comprising two core components: a trigger mechanism and an intervention process. This mechanism effectively helps models distinguish between objective world states and subjective belief states, significantly improving social reasoning accuracy and consistency on three social reasoning benchmarks ToMi, Hi-ToM, and ExploreToM. Additionally, some limitations and potential future works are listed as follows:

- **Trigger Mechanism Research** Our current trigger mechanism relies on predefined intervention words, potentially missing other forms of cognitive confusion expressions. Future research should develop adaptive trigger mechanisms to automatically identify confusion states without predefined word lists.

- **Experiments in Multiple LLMs and Generalizability** Our experiments primarily focus on theory-of-mind benchmarks and DeepSeek-R1 series LLMs. Future work may conduct extensive evaluations across different reasoning LLMs to establish thorough intervention strategies and expand to broader social reasoning tasks such as moral reasoning or social norm understanding.

ETHICS STATEMENT

Our research follows ethical guidelines for artificial intelligence research and only uses publicly available academic benchmark datasets (ToMi, Hi-ToM, and ExploreToM), involving no human subjects. Our analysis of DeepSeek-R1's reasoning trajectories is purely for scientific research purposes, aimed at improving social reasoning capabilities of LLMs. Our use of LLMs is entirely reasonable and complies with academic research standards, with the paper containing no personal privacy information, emotional manipulation content, or biased materials. All authors of this research declare no conflicts of interest regarding this submission, and we openly share our research methodology and clearly disclose the limitations of our approach to encourage deeper future research. All procedures and methods in this research comply with ICLR's ethical standards and academic publication requirements.

REPRODUCIBILITY STATEMENT

We have made extensive efforts to ensure the reproducibility of our work. The datasets used for our experiment are explicitly described in Appendix A.2. For the methodology, we provide complete world model design and implementation procedure. Maximum token length, temperature and other parameters, are thoroughly documented in the paper. In the supplementary materials, we also submit the source code and datasets required to reproduce our method.

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

## A APPENDIX

### A.1 THE USE OF LARGE LANGUAGE MODELS (LLMS)

In compliance with the ICLR 2026 disclosure requirements on language model usage, we confirm that the use of LLMs in this study was strictly limited to linguistic refinement. Specifically, they were employed to improve syntactic structure, enhance academic style, standardize terminology, and increase the readability of technical content, thereby facilitating clearer scientific communication. Importantly, LLMs were not involved in generating research ideas, designing methodologies, or contributing to scientific conclusions; these aspects were carried out solely by the authors.

## A.2 DATASETS

**ToMi** The dataset is a social reasoning dataset that describes characters' social states in simple scenarios, containing extensive scene descriptions and character belief state descriptions.

**HiToM** The scene descriptions and character belief state descriptions are more complex, involving deeper cognitive processes and higher-order reasoning.

**ExploreToMi** The dataset is more aligned with realistic scenarios, featuring more diverse character behaviors and incorporating scene descriptions such as weather conditions, atmospheric changes, and other objective influencing factors.

## A.3 CASE STUDY

In the Figure 6, we present a complete reasoning case to demonstrate how the model's reasoning becomes trapped in confusion and how our method achieves the correct answer while significantly reducing token consumption. Meanwhile, the tokens required to construct the textual world model are minimal, with essentially no impact on the total reasoning token consumption, so we include them in our token count statistics.

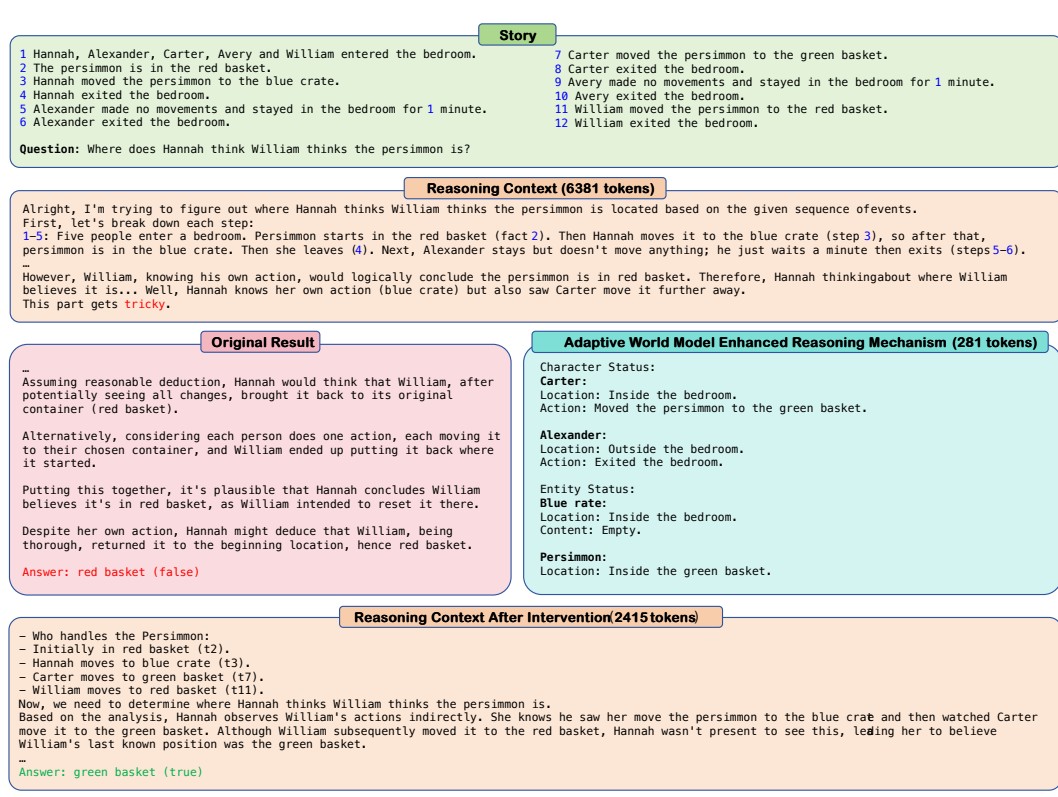

Figure 6: Samples of reasoning trajectories with and without adaptive world model-enhanced reasoning mechanism

## A.4 SAMPLE EXAMPLES FROM DATASET

For the data sources we use, we present sample examples from Figure 7 to Figure 9. Figure 7 describes characters' social states in simple scenarios, containing extensive scene descriptions and character belief state descriptions. Figure 8 presents a sample example from the HiToM dataset, where the Story consists of multiple and complex descriptions, with its deeper cognitive processes and higher-order reasoning. Figure 9 presents a sample example from the ExploreToM dataset, whose Story is more aligned with realistic scenarios, including some advanced socio-cognitive events, such as "told privately", "in secret", "got distracted", etc.

```
                                    ToMi
        Story:
        1 Isla entered the porch.
        2 Isabella entered the living_room.
        3 Jack entered the porch.
        4 Jack loves the strawberry.
        5 The tangerine is in the basket.
        6 The basket is in the porch.
        7 Isla loves the trousers.
        8 Jack moved the tangerine to the suitcase.
        9 The suitcase is in the porch.
        10 Isla exited the porch.
        11 Jack exited the porch.
        12 Isla entered the porch.

        Question:
        Where does Isla think that Jack searches for the tangerine?

        Choices:
        basket, suitcase
```

Figure 7: Sample example from dataset ToMi

```
                                    Hi-ToM
Story:
1 Hannah, Alexander, Carter, Avery and William entered the bedroom.
2 The persimmon is in the red_basket.
3 Hannah moved the persimmon to the blue_crate.
4 Hannah exited the bedroom.
5 Alexander made no movements and stayed in the bedroom for 1 minute.
6 Alexander exited the bedroom.
7 Carter moved the persimmon to the green_basket.
8 Carter exited the bedroom.
9 Avery made no movements and stayed in the bedroom for 1 minute.
10 Avery exited the bedroom.
11 William moved the persimmon to the red_basket.
12 William exited the bedroom.
13 Hannah, Alexander, Carter, Avery and William entered the waiting_room.
14 William, Alexander and Hannah entered the bathroom.
15 The banana is in the red_bucket.
16 William moved the banana to the red_crate.
17 William exited the bathroom.
18 Alexander made no movements and stayed in the bathroom for 1 minute.
19 Alexander exited the bathroom.
20 Hannah made no movements and stayed in the bathroom for 1 minute.
21 Hannah exited the bathroom.
22 William, Alexander and Hannah entered the waiting_room.
23 Alexander, Avery, William and Hannah entered the garage.
24 The plum is in the red_drawer.
25 Alexander moved the plum to the green_cupboard.
26 Alexander exited the garage.
27 Avery moved the plum to the blue_treasure_chest.

Question:
Where does Hannah think William thinks the persimmon is?

Choices:
red_crate,  red_bucket,  red_box,  blue_bucket,  green_box,  red_basket,  blue_crate,  green_bathtub,
green_envelope, green_basket, red_drawer, green_bottle, blue_bathtub, blue_treasure_chest, green_cupboard.
```

Figure 8: Sample example from dataset Hi-ToM

**ExploreToM**

Story:
1 Lucas entered the festival merchandise booth.
2 Lucas moved the portable speaker to the duffel bag, which is also located in the festival merchandise booth.
3 Lucas moved the portable speaker to the main information tent in secret, leaving the duffel bag in its original location.
4 Lucas entered the festival merchandise booth.
5 Lucas left the festival merchandise booth.
6 Lucas told privately to Danielle that the duffel bag is in the festival merchandise booth.
7 Danielle told privately to Alexis that the duffel bag is in the festival merchandise booth.
8 Danielle entered the main information tent.
9 Lucas told privately to Danielle that the portable speaker is in the main information tent.
10 Lucas told privately to Alexis that the portable speaker is in the main information tent.
11 Lucas told privately to Alexis that the duffel bag is in the festival merchandise booth.
12 Danielle moved the portable speaker to the duffel bag, while Lucas got distracted and did not realize.

Question:
In which room does Alexis think that Lucas will search for the duffel bag?

Figure 9: Sample example from dataset ExploreToM

## A.5 SAMPLE EXAMPLE OF WORLD MODEL GENERATION

To better illustrate the effectiveness of adaptive world model-enhanced reasoning mechanism, we present a comprehensive set of examples demonstrating textual world model generation, as illustrated in the Figure 10. The world model maintains dynamic tracking of character and entity states, which are continuously updated in response to different actions and event. This dynamic updating capability enables our method to flexibly adapt to diverse story scenarios, ensuring that during the intervention process, the system can provide accurate information corresponding to the current state of the narrative.

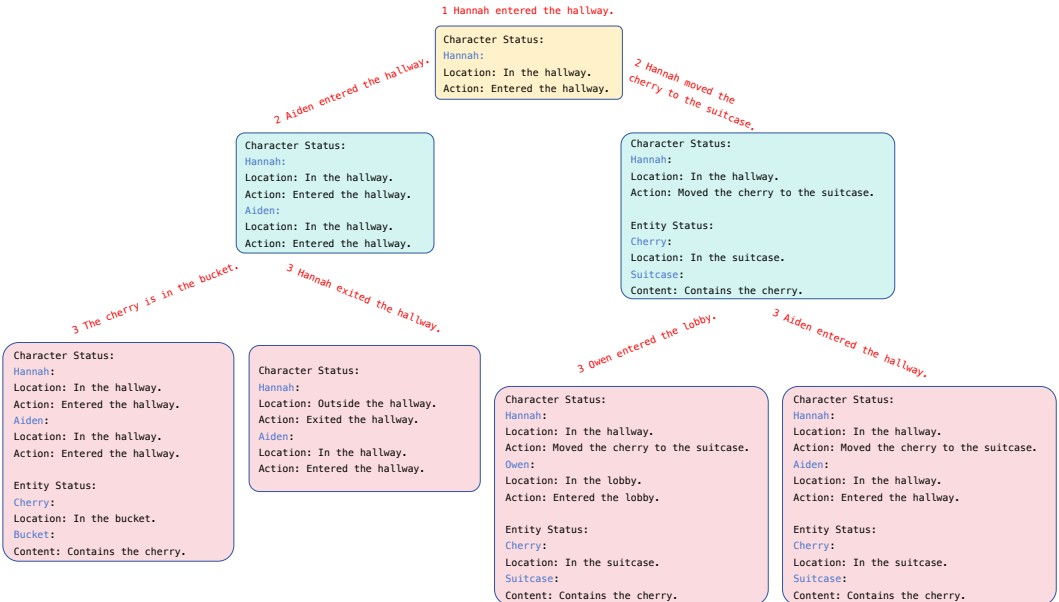

Figure 10: Sample example of World Model Generation

