# OpenReview forum: "Active Confusion Expression in Large Language Models: Leveraging World Models toward Better Social Reasoning"
_ICLR.cc/2026/Conference — ICLR 2026 Conference Withdrawn Submission_

### Official Review · Reviewer_rpMK · 2025-10-25

**Soundness:** 3
**Presentation:** 3
**Contribution:** 3
**Rating:** 4
**Confidence:** 4

**Summary:**

The paper proposes an Active Confusion Expression with Adaptive World Model framework to improve LLMs’ social reasoning.
It detects moments when a model shows uncertainty (e.g., “I’m confused”) and dynamically injects a structured “world state” and “belief state” clarification to guide reasoning.
Experiments on three Theory-of-Mind benchmarks demonstrate consistent accuracy and efficiency improvements over existing reasoning methods.

**Strengths:**

Novel and intuitive idea.
Using confusion signals as triggers for reasoning intervention is creative and aligns with human metacognitive behavior.

Empirical effectiveness.
The approach improves accuracy and reduces token usage across multiple ToM datasets.

Interpretability.
The intervention makes reasoning steps more explicit and easier to analyze.

**Weaknesses:**

The main limitation of this paper lies in its lack of theoretical rigor and generalization analysis. While the distinction between “world state” and “belief state” is conceptually appealing, it remains informally defined without a formal cognitive or probabilistic framework. The confusion detection mechanism relies on a fixed list of trigger words, making it heuristic and potentially fragile across languages, models, and task domains. Moreover, the paper does not analyze how the adaptive world model intervention changes the model’s internal reasoning dynamics, leaving the mechanism’s effectiveness somewhat opaque. Experimental validation is restricted to one primary model (DeepSeek-R1-Distill-Qwen-32B) and a few social reasoning benchmarks, with no exploration of broader reasoning contexts.

**Questions:**

See weaknesses.

---

> ### Author Response · Authors · 2025-11-26
> **Rebuttal for W1**
>
> Dear Reviewers,
>
> We sincerely thank for your thoughtful and constructive feedback. We have carefully considered each comment and provide detailed responses below. We believe these clarifications and additional experiments significantly strengthen our work.
>
> ## Rebuttal for W1
>
> **Lack of theoretical rigor and generalization analysis.**
>
> ---
>
> **Cognitive Science Theoretical Foundation:** Our method has solid **cognitive science theoretical support**:
>
> 1. **Representation-Reality Distinction**: This is a core capability in Baron-Cohen's theory of mind development model, referring to an individual's ability to distinguish between "the objective state of the world" and "others' subjective representations of the world"
>
> 2. **False Belief Understanding**: Classic studies by Premack & Woodruff (1978) and Wimmer & Perner (1983) established this distinction as the essence of ToM—understanding that others' beliefs may not align with reality
>
> 3. **Observability and Belief Formation**: Our world model records objective facts while belief state tracks observation-based beliefs, directly corresponding to the "seeing-leads-to-knowing" principle in cognitive development research
>
> **Value of Empirical Approach:** Our work adopts an **empirically-driven methodology**:
> - **Operationalizes** cognitive science theory into computable mechanisms (structured state representation and dynamic updates)
> - **Systematically validates** this cognitive principle's effectiveness in LLMs through three standard ToM benchmarks
> - Provides **reproducible evidence** of 3-7% accuracy improvement and 18-34% token reduction

---

> ### Author Response · Authors · 2025-11-26
> **Rebuttal for W2**
>
> ## Rebuttal for W2
>
> **The confusion detection mechanism relies on a fixed list of trigger words.**
>
> ---
>
> ### Empirical Effectiveness: High Coverage and Significant Improvements
>
> Our intervention word trigger mechanism demonstrates consistent effectiveness across three benchmarks:
>
> - **ToMi**: Intervention covers 48% of tasks, accuracy improves from 79.33% to 83.67% (+4.34%), tokens reduce from 3504 to 2585 (-26.23%)
> - **HiToM**: Intervention covers 59% of tasks, accuracy improves from 69.33% to 76.67% (+7.34%), tokens reduce from 7541 to 4993 (-33.79%)
> - **ExploreTom**: Intervention covers 43% of tasks, accuracy improves from 69.67% to 72.67% (+3.00%), tokens reduce from 3185 to 2609 (-18.08%)
>
> These results demonstrate that despite theoretical brittleness, intervention word triggering effectively identifies critical moments requiring intervention in practice.
>
> ### Comparison with Alternative Methods: Avoiding Over-Intervention
>
> We tested **confidence-based** and **entropy-based** dynamic triggering methods and found that these seemingly more sophisticated approaches suffer from severe over-intervention problems:
>
> **Issue 1 - False Trigger Noise:** Uncertainty-based methods frequently trigger interventions on irrelevant tokens, such as:
> - Proper names (e.g., "Andy")
> - Pronouns (e.g., "he", "she")
> - Common nouns (e.g., "table", "box")
> - Reasoning markers (e.g., "let me think")
>
> **Issue 2 - Semantic Misjudgment:** These methods cannot distinguish between normal thinking pauses during reasoning and genuine cognitive confusion. When a model outputs "let me think," it does not necessarily indicate confusion—it may simply be part of its reasoning style.
>
> **Issue 3 - Premature Saturation:** Due to high false trigger rates, intervention thresholds are easily reached regardless of task complexity, leading to:
> - Unnecessary token waste on simple tasks
> - Premature exhaustion of intervention budget on complex tasks, preventing help when truly needed
> - Minimal accuracy improvement, potentially even degradation due to over-intervention
>
> To validate the advantages of our confusion word trigger mechanism over dynamic uncertainty-based methods, we conducted comparative experiments on DeepSeek-R1-Distill-Qwen-32B.
>
> ### Experimental Setup
>
> **Confidence-Based Method:**
> - Monitors output probability of each token
> - Threshold: Triggers intervention when confidence < 0.3
> - Rationale: Low confidence indicates model uncertainty
>
> **Entropy-Based Method:**
> - Calculates entropy of output distribution (averaged over 5 samples)
> - Threshold: Triggers intervention when entropy > 1.5
> - Rationale: High entropy indicates output instability
>
> **Our Method:**
> - Triggers based on confusion word language signals
> - Intervenes only at explicit cognitive confusion expressions
>
> ### Comparative Results
>
> The following table shows performance comparison of three methods across three benchmarks:
>
> | Method | ToMi |  | HiToM |  | ExploreTom |  |
> |--------|------|------|-------|------|------------|------|
> |  | Acc | Token | Acc | Token | Acc | Token |
> | **Baseline** | 79.33 | 3504 | 69.33 | 7541 | 69.67 | 3185 |
> | **Confidence** | 81.00 | 5127 | 72.67 | 8145 | 70.00 | 4803 |
> | Change | +1.67% | **+46.3%**  | +3.34% | **+8.0%**  | +0.33% | **+50.8%**  |
> | **Entropy** | 81.67 | 4839 | 74.00 | 8072 | 70.67 | 4960 |
> | Change | +2.34% | **+38.1%**  | +4.67% | **+7.0%**  | +1.00% | **+55.8%**  |
> | **Ours** | 83.67 | 2585 | 76.67 | 4993 | 72.67 | 2609 |
> | Change | **+4.34%**  | **-26.2%**  | **+7.34%**  | **-33.8%**  | **+3.00%**  | **-18.1%**  |
>
> ### Key Results
>
> **1. Over-Intervention Problem:** Confidence and entropy methods reach the maximum intervention threshold (k=3) in almost all tasks, indicating they trigger frequent interventions, including many unnecessary ones.
>
> **2. Limited Accuracy Improvement:**
> - Confidence method: Average improvement only 1.78%
> - Entropy method: Average improvement 2.67%
> - **Our method: Average improvement 4.89% (2.7× entropy method)**
>
> **3. Severe Token Waste:** Confidence and entropy methods not only fail to reduce token consumption but cause **significant increases** (averaging +8% to +56%), due to:
> - Frequent triggering at irrelevant positions (names, pronouns, common nouns)
> - False triggers during normal reasoning (e.g., "let me think")
> - Quick threshold saturation in both simple and complex tasks, preventing targeted intervention budget allocation
>
> **4. Advantages of Our Method:**
> - **Selective intervention**: Triggers only at genuine cognitive confusion
> - **Highest accuracy**: Achieves maximum improvement across all three datasets
> - **High efficiency**: Averages 26% token reduction

---

> > ### Author Response · Authors · 2025-11-26
> > **Rebuttal for W2(continued)**
> >
> > ### Concrete Case Analysis
> >
> > The following case demonstrates the false trigger problem of the entropy-based method:
> >
> > ```
> > Okay, so I'm trying to figure out where Owen thinks Abigail would look for the socks based on the given statements. Let me go through each step one by one.
> >
> > First, let's list all the points provided:
> >
> > 1. Owen doesn't like broccoli. Hmm, probably irrelevant here but maybe his dislike affects his actions or thoughts somehow? Not sure yet.
> > 2. Owen enters the cellar. So he was there at some point before.
> > 3. Abigail also enters the cellar after him. She joins him down there.
> > 4. There's a sock inside an envelope. That seems important because we're dealing with where things are placed.
> > 5. This envelope is located in the cellar. So initially, both the envelope and the socks were in the same place as Owen and Abigail when they went into the cellar together.
> > 6. Then Abigail moves the socks from the envelope (which was in the cellar) to the cupboard. Wait, so she took them out of the envelope and put them somewhere else within the same location—the cellar—since the next statement says...
> > 7. ...the cupboard is indeed in the cellar. Oh, okay, so moving to another spot still keeps everything underground in the same room basically.
> > 8. After these movements, Owen exits the cellar. He leaves while the others might not have.
> >
> > Now, thinking about what Owen knows up until his exit. When Owen left, he knew the socks had been moved to the cupboard in the cellar by Abigail. But since Owen himself didn't move anything—he just came and left—it depends on whether he observed her action or if he only saw their initial positions.
> >
> > Wait, did Owen see Abigail move the socks? Or did he leave before she made the change? Looking back, the order is: Owen goes in ...
> >
> > in：High Entropy Trigger
> >
> > ```
> >
> > **Problem Explanation:** The entropy method detects high entropy at common prepositions like "in" (possibly because various nouns can follow), incorrectly triggering intervention. However, this is not where the model encounters ToM confusion. Such false triggers lead to:
> > 1. Premature consumption of intervention budget
> > 2. World model injection where help is not needed, causing reasoning interruption
> > 3. Token budget waste without solving real reasoning difficulties
> >
> > In contrast, our confusion word method triggers only when the model explicitly expresses confusion (e.g., "confused about", "unclear whether"), ensuring each intervention targets a genuine cognitive bottleneck.

---

> ### Author Response · Authors · 2025-11-26
> **Rebuttal for W3**
>
> ## Rebuttal for W3
>
> **How the adaptive world model intervention changes the model’s internal reasoning dynamics.**
>
> ---
>
> **Locality and Dynamics of World Model:** Our world model only records the objective factual states of characters and entities **at the current reasoning stage**, rather than a globally consistent complete world state. The world model is **dynamically updated** as the story unfolds, reflecting changes in objective facts at different time points, thus avoiding the "omniscient perspective" problem of global consistency.
>
> **Strict Fact-Belief Separation:** Critically, our world model **never directly provides agent belief or cognitive state information** (e.g., "X believes Y is at location Z"). The world model only contains objective facts (e.g., "Y is actually at location Z"), and the model still needs to reason about:
> - Which agents observed these facts
> - What beliefs agents would form based on their observations
> - The discrepancies between agent beliefs and objective facts
>
> **Preserving ToM Reasoning Integrity:** Therefore, our method does not short-circuit the core ToM skill—the ability to distinguish facts from beliefs. On the contrary, by providing a clear objective fact baseline, the model can more accurately perform the "fact-observation-belief" reasoning chain, which is the essence of ToM.
>
> **Illustrative Case:** The following case demonstrates how the world model maintains fact-belief separation:
>
> ```
> Story:
> 1 Emma, Lucas, Sophia and Noah entered the kitchen.
> 2 The cake is in the white box.
> 3 Emma moved the cake to the blue container.
> 4 Emma exited the kitchen.
> 5 Lucas made no movements and stayed in the kitchen for 1 minute.
> 5 Lucas exited the kitchen.
> 6 Sophia moved the cake to the red tin.
> 7 Sophia exited the kitchen.
> 8 Noah moved the cake to the white box.
> 9 Noah exited the kitchen.
> Question: Where does Emma think Noah thinks the cake is?
>
> *** Reasoning Context: ***
> Alright, I'm trying to figure out where Emma thinks Noah thinks the cake is located based on the given sequence of events. Let's break down each step:
> Steps 1-3: Four people enter the kitchen. The cake starts in the white box (fact 2). Then Emma moves it to the blue container (step 3), so after that, the cake is in the blue container. Then she leaves (step 4). Next, Lucas stays but doesn't move anything; he just waits a minute then exits (steps 5-6).
> However, Noah, knowing his own action, would logically conclude the cake is in the white box (since he moved it there himself in step 9). Therefore, Emma thinking about where Noah believes it to be needs to account for what Noah saw and what Emma knows about Noah's perspective. This part gets tricky.
>
> World Model:
> Character Status:
> Emma:
> Location: Outside the kitchen. Action: Moved the cake to the blue container.
> Lucas:
> Location: Outside the kitchen. Action: Exited the kitchen.
> Noah:
> Location: Inside the kitchen. Action: No movements yet.
> Sophia:
> Location: Inside the kitchen. Action: No movements yet.
> Entity Status:
> White box:
> Location: Inside the kitchen. Content: Empty.
> Blue container:
> Location: Inside the kitchen. Content: Cake.
> Red tin:
> Location: Inside the kitchen. Content: Empty.
> Cake:
> Location: Inside the blue container.
>
> Reasoning Context After Intervention:
> Tracking the cake's journey:
> Initially in white box (step 2)
> Emma moves to blue container (step 3)
> Sophia later moves to red tin (step 7)
> Noah finally moves to white box (step 9)
> Now, we need to determine where Emma thinks Noah thinks the cake is.
> Based on the world model analysis, Emma's observational window closed at step 4 when she exited the kitchen. At that moment, she had just moved the cake to the blue container. Critically, Noah was still present in the kitchen and was positioned to witness Emma's action directly.
> From Emma's perspective, she knows:
> 1.She moved the cake to the blue container
> 2.Noah was present and observed her doing this
> Emma's reasoning about Noah's belief state would therefore be: Noah saw me move the cake to the blue container. Since I left immediately after, the last thing Noah observed regarding the cake's location was my placement of it in the blue container. Emma has no reason to suspect Noah performed any action, as she simply exited after her own movement.
> The key insight is that Emma believes Noah's knowledge stopped at the same point hers did - when the cake was in the blue container. She would think Noah, having witnessed her action, believes the cake remains in the blue container. Emma doesn't know that Noah later moved it, nor that Sophia moved it before Noah.
> Therefore, Emma thinks Noah thinks the cake is in the blue container.Emma projects her last-known state (cake in blue container) onto what she imagines Noah believes, since Noah was present to witness that final state from Emma's perspective.
> Answer: blue container (correct)
> ```

---

> ### Author Response · Authors · 2025-11-26
> **Rebuttal for W4**
>
> ## Rebuttal for W4
>
> **Experimental validation is restricted to one primary model**
>
> ---
>
> ### Multi-Model Evaluation
>
> **Closed-Source Frontier Model Baselines:** We have reported the performance of multiple frontier models across three benchmarks in Table 2:
> - Complete baseline results for **GPT-4o, GPT-5, and Claude-Sonnet-4**
> - These results show that even the strongest models still have room for improvement on ToM tasks
>
> **In-Depth Open-Source Model Evaluation:** Our primary experiments were conducted on **DeepSeek-R1-Distill-Qwen-32B**, a representative open-source reasoning model. To validate our method's effectiveness across different capability levels, we have supplemented results for **DeepSeek-R1** (the full version with stronger reasoning capabilities):
>
> | Model | ToMi (Acc/Token) | HiToM (Acc/Token) | ExploreTom (Acc/Token) |
> |-------|------------------|-------------------|----------------------|
> | **DeepSeek-R1-Distill-Qwen-32B** | 79.33 / 3504 | 69.33 / 7541 | 69.67 / 3185 |
> | + Our Method | **83.67 / 2585** | **76.67 / 4993** | **72.67 / 2609** |
> | Improvement | +4.34% / -26.2% | +7.34% / -33.8% | +3.00% / -18.1% |
> | **DeepSeek-R1** | 93.33 / 2846 | 77.33 / 7710 | 78.00 / 2124 |
> | + Our Method | **95.67 / 2119** | **87.67 / 4217** | **83.33 / 1362** |
> | Improvement | +2.34% / -25.5% | +10.34% / -45.3% | +5.33% / -35.9% |
>
> Experimental results demonstrate our method achieves significant improvements in both accuracy and token efficiency.
>
> ### Rationale for Domain-Specific Design
>
> **Necessity of ToM Focus:** Our method is a **domain-specific design for ToM**, not a general reasoning framework:
> - **Design principles** (world-belief separation, confusion word triggering) are optimized specifically for social reasoning
> - **Cognitive mechanism** (cognitive duality modeling) directly corresponds to ToM's core capabilities
> - This focus enables us to deeply address ToM-specific challenges
>
> **No Negative Transfer Risk:** As a **training-free prompting method**, our approach:
> - Does not modify model parameters or introduce extraneous structures
> - Has no parameter-level negative transfer or catastrophic forgetting risk
> - When ToM-specific prompting is not applied, model behavior remains identical to baseline

---

### Official Review · Reviewer_XFub · 2025-10-25

**Soundness:** 3
**Presentation:** 2
**Contribution:** 2
**Rating:** 4
**Confidence:** 4

**Summary:**

This paper investigates the phenomenon of cognitive confusion in social reasoning within LLMs. While LLMs have achieved strong performance in mathematical and code reasoning, the authors show that they often fail at theory-of-mind tasks that require distinguishing objective world states from subjective belief states.
The central contribution is an adaptive world-model–enhanced reasoning mechanism that detects “confusion” during reasoning and injects structured world-state information to help the model recover coherent reasoning trajectories. The method improves accuracy on three social reasoning benchmarks — ToMi, Hi-ToM, and Explore-ToM.

**Strengths:**

1. Novel diagnosis of reasoning breakdowns. The paper introduces Active Confusion Expression, which is an observable linguistic signal of reasoning failure.

2. Detailed semantic similarity analysis. The paper conducts a Semantic Similarity Analysis of Context-Disrupting Words.
For each word in DeepSeek-R1’s reasoning trajectories, they compute cosine similarity between the surrounding context embeddings.

**Weaknesses:**

1. Limited applicability to non-reasoning models.

2. Heuristic trigger detection.

3. Insufficient methodological details.

**Questions:**

1. The method relies on detecting confusion during explicit reasoning traces (e.g., CoT). Non-reasoning models, which generate short direct outputs without introspective tokens, do not provide a trajectory where intervention can occur. Would you explain whether the method can be applied to a non-reasoning model?

2. Although the paper employs Semantic Similarity Analysis offline to identify confusion words, the runtime detection mechanism is limited to simple keyword matching rather than continuous semantic monitoring. I wonder if the authors can compare their approach with alternative runtime intervention strategies, such as confidence-based examination or other uncertainty-aware methods, to better assess the effectiveness of their trigger design.

3. The process for constructing the textual world model is under-specified. The paper briefly states that it “tracks entity and character states,” but does not explain how these states are parsed or updated. In addition, the paper does not provide enough information about how the Semantic Similarity Analysis of Context-Disrupting Words was implemented.

---

> ### Author Response · Authors · 2025-11-26
> **Rebuttal for W1&Q1**
>
> Dear Reviewers,
>
> We sincerely thank for your thoughtful and constructive feedback. We have carefully considered each comment and provide detailed responses below. We believe these clarifications and additional experiments significantly strengthen our work.
>
> ## Rebuttal for W1&Q1
>
> **Whether the method can be applied to a non-reasoning model.**
>
> ---
>
> Our method is **not applicable to non-reasoning models** because it relies on detecting language signals during the reasoning process to trigger interventions. Non-reasoning models lack explicit reasoning traces and cannot provide the intermediate outputs required for confusion detection.
>
> However, if future model developments can provide internal reasoning traces of non-reasoning models (e.g., through supervised fine-tuning or chain-of-thought distillation to obtain implicit reasoning steps), our method could in principle be extended to these scenarios.

---

> ### Author Response · Authors · 2025-11-26
> **Rebuttal for W2&Q2**
>
> ## Rebuttal for W2&Q2
>
> **Heuristic trigger detection.**
>
> ---
>
> ### Empirical Effectiveness: High Coverage and Significant Improvements
>
> Our intervention word trigger mechanism demonstrates consistent effectiveness across three benchmarks:
>
> - **ToMi**: Intervention covers 48% of tasks, accuracy improves from 79.33% to 83.67% (+4.34%), tokens reduce from 3504 to 2585 (-26.23%)
> - **HiToM**: Intervention covers 59% of tasks, accuracy improves from 69.33% to 76.67% (+7.34%), tokens reduce from 7541 to 4993 (-33.79%)
> - **ExploreTom**: Intervention covers 43% of tasks, accuracy improves from 69.67% to 72.67% (+3.00%), tokens reduce from 3185 to 2609 (-18.08%)
>
> These results demonstrate that despite theoretical brittleness, intervention word triggering effectively identifies critical moments requiring intervention in practice.
>
> ### Comparison with Alternative Methods: Avoiding Over-Intervention
>
> We tested **confidence-based** and **entropy-based** dynamic triggering methods and found that these seemingly more sophisticated approaches suffer from severe over-intervention problems:
>
> **Issue 1 - False Trigger Noise:** Uncertainty-based methods frequently trigger interventions on irrelevant tokens, such as:
> - Proper names (e.g., "Andy")
> - Pronouns (e.g., "he", "she")
> - Common nouns (e.g., "table", "box")
> - Reasoning markers (e.g., "let me think")
>
> **Issue 2 - Semantic Misjudgment:** These methods cannot distinguish between normal thinking pauses during reasoning and genuine cognitive confusion. When a model outputs "let me think," it does not necessarily indicate confusion—it may simply be part of its reasoning style.
>
> **Issue 3 - Premature Saturation:** Due to high false trigger rates, intervention thresholds are easily reached regardless of task complexity, leading to:
> - Unnecessary token waste on simple tasks
> - Premature exhaustion of intervention budget on complex tasks, preventing help when truly needed
> - Minimal accuracy improvement, potentially even degradation due to over-intervention
>
> To validate the advantages of our confusion word trigger mechanism over dynamic uncertainty-based methods, we conducted comparative experiments on DeepSeek-R1-Distill-Qwen-32B.
>
> ### Experimental Setup
>
> **Confidence-Based Method:**
> - Monitors output probability of each token
> - Threshold: Triggers intervention when confidence < 0.3
> - Rationale: Low confidence indicates model uncertainty
>
> **Entropy-Based Method:**
> - Calculates entropy of output distribution (averaged over 5 samples)
> - Threshold: Triggers intervention when entropy > 1.5
> - Rationale: High entropy indicates output instability
>
> **Our Method:**
> - Triggers based on confusion word language signals
> - Intervenes only at explicit cognitive confusion expressions
>
> ### Comparative Results
>
> The following table shows performance comparison of three methods across three benchmarks:
>
> | Method | ToMi |  | HiToM |  | ExploreTom |  |
> |--------|------|------|-------|------|------------|------|
> |  | Acc | Token | Acc | Token | Acc | Token |
> | **Baseline** | 79.33 | 3504 | 69.33 | 7541 | 69.67 | 3185 |
> | **Confidence** | 81.00 | 5127 | 72.67 | 8145 | 70.00 | 4803 |
> | Change | +1.67% | **+46.3%**  | +3.34% | **+8.0%**  | +0.33% | **+50.8%**  |
> | **Entropy** | 81.67 | 4839 | 74.00 | 8072 | 70.67 | 4960 |
> | Change | +2.34% | **+38.1%**  | +4.67% | **+7.0%**  | +1.00% | **+55.8%**  |
> | **Ours** | 83.67 | 2585 | 76.67 | 4993 | 72.67 | 2609 |
> | Change | **+4.34%**  | **-26.2%**  | **+7.34%**  | **-33.8%**  | **+3.00%**  | **-18.1%**  |
>
> ### Key Results
>
> **1. Over-Intervention Problem:** Confidence and entropy methods reach the maximum intervention threshold (k=3) in almost all tasks, indicating they trigger frequent interventions, including many unnecessary ones.
>
> **2. Limited Accuracy Improvement:**
> - Confidence method: Average improvement only 1.78%
> - Entropy method: Average improvement 2.67%
> - **Our method: Average improvement 4.89% (2.7× entropy method)**
>
> **3. Severe Token Waste:** Confidence and entropy methods not only fail to reduce token consumption but cause **significant increases** (averaging +8% to +56%), due to:
> - Frequent triggering at irrelevant positions (names, pronouns, common nouns)
> - False triggers during normal reasoning (e.g., "let me think")
> - Quick threshold saturation in both simple and complex tasks, preventing targeted intervention budget allocation
>
> **4. Advantages of Our Method:**
> - **Selective intervention**: Triggers only at genuine cognitive confusion
> - **Highest accuracy**: Achieves maximum improvement across all three datasets
> - **High efficiency**: Averages 26% token reduction

---

> > ### Author Response · Authors · 2025-11-26
> > **Rebuttal for W2&Q2(continued)**
> >
> > #### Concrete Case Analysis
> >
> > The following case demonstrates the false trigger problem of the entropy-based method:
> >
> > ```
> > Okay, so I'm trying to figure out where Owen thinks Abigail would look for the socks based on the given statements. Let me go through each step one by one.
> >
> > First, let's list all the points provided:
> >
> > 1. Owen doesn't like broccoli. Hmm, probably irrelevant here but maybe his dislike affects his actions or thoughts somehow? Not sure yet.
> > 2. Owen enters the cellar. So he was there at some point before.
> > 3. Abigail also enters the cellar after him. She joins him down there.
> > 4. There's a sock inside an envelope. That seems important because we're dealing with where things are placed.
> > 5. This envelope is located in the cellar. So initially, both the envelope and the socks were in the same place as Owen and Abigail when they went into the cellar together.
> > 6. Then Abigail moves the socks from the envelope (which was in the cellar) to the cupboard. Wait, so she took them out of the envelope and put them somewhere else within the same location—the cellar—since the next statement says...
> > 7. ...the cupboard is indeed in the cellar. Oh, okay, so moving to another spot still keeps everything underground in the same room basically.
> > 8. After these movements, Owen exits the cellar. He leaves while the others might not have.
> >
> > Now, thinking about what Owen knows up until his exit. When Owen left, he knew the socks had been moved to the cupboard in the cellar by Abigail. But since Owen himself didn't move anything—he just came and left—it depends on whether he observed her action or if he only saw their initial positions.
> >
> > Wait, did Owen see Abigail move the socks? Or did he leave before she made the change? Looking back, the order is: Owen goes in ...
> >
> > in：High Entropy Trigger
> >
> > ```
> >
> > **Problem Explanation:** The entropy method detects high entropy at common prepositions like "in" (possibly because various nouns can follow), incorrectly triggering intervention. However, this is not where the model encounters ToM confusion. Such false triggers lead to:
> > 1. Premature consumption of intervention budget
> > 2. World model injection where help is not needed, causing reasoning interruption
> > 3. Token budget waste without solving real reasoning difficulties
> >
> > In contrast, our confusion word method triggers only when the model explicitly expresses confusion (e.g., "confused about", "unclear whether"), ensuring each intervention targets a genuine cognitive bottleneck.

---

> ### Author Response · Authors · 2025-11-26
> **Rebuttal for W3&Q3**
>
> ## Rebuttal for W3&Q3
>
> **Insufficient methodological details.**
>
> ---
>
> ### World Model Construction Process
>
> **Structured State Extraction and Update:** We use structured prompts to guide the model in progressively extracting and updating entity states from the story:
>
> **State Tracking Content:**
> - **Entity Location**: Current location of each entity
> - **Entity Action**: Key actions performed by entities and their results
> - **Character Relationships**: Interactions between characters and observability
>
> **Update Mechanism:** World model is **dynamically updated** as the story unfolds:
> 1. Initialization: Extract entities and locations at story start
> 2. Incremental update: When new events occur, only update affected entity states
> 3. Concise representation: Generate structured text summaries avoiding redundancy
>
> **Concrete Example:**
>
> ```
> Story:
> 1 Emma, Lucas, Sophia and Noah entered the kitchen.
> 2 The cake is in the white box.
> 3 Emma moved the cake to the blue container.
> 4 Emma exited the kitchen.
> 5 Lucas made no movements and stayed in the kitchen for 1 minute.
> 6 Lucas exited the kitchen.
> 7 Sophia moved the cake to the red tin.
> 8 Sophia exited the kitchen.
> 9 Noah moved the cake to the white box.
> 10 Noah exited the kitchen.
>
> Step-by-Step Construction Process:
>
> 1 Emma, Lucas, Sophia and Noah entered the kitchen.
>
> Character Status:
> Emma:
> Location: Inside the kitchen. Action: Entered the kitchen.
> Lucas:
> Location: Inside the kitchen. Action: Entered the kitchen.
> Sophia:
> Location: Inside the kitchen. Action: Entered the kitchen.
> Noah:
> Location: Inside the kitchen. Action: Entered the kitchen.
>
> 2 The cake is in the white box.
>
> Character Status:
> Emma:
> Location: Inside the kitchen. Action: Entered the kitchen.
> Lucas:
> Location: Inside the kitchen. Action: Entered the kitchen.
> Sophia:
> Location: Inside the kitchen. Action: Entered the kitchen.
> Noah:
> Location: Inside the kitchen. Action: Entered the kitchen.
> Entity Status:
> White box:
> Location: Inside the kitchen. Content: Cake.
> Cake:
> Location: Inside the white box.
>
> 3 Emma moved the cake to the blue container.
>
> Character Status:
> Emma:
> Location: Inside the kitchen. Action: Moved the cake to the blue container.
> Lucas:
> Location: Inside the kitchen. Action: Entered the kitchen.
> Sophia:
> Location: Inside the kitchen. Action: Entered the kitchen.
> Noah:
> Location: Inside the kitchen. Action: Entered the kitchen.
> Entity Status:
> White box:
> Location: Inside the kitchen. Content: Empty.
> Blue container:
> Location: Inside the kitchen. Content: Cake.
> Cake:
> Location: Inside the blue container.
>
> 4 Emma exited the kitchen.
>
> Character Status:
> Emma:
> Location: Outside the kitchen. Action: Exited the kitchen.
> Lucas:
> Location: Inside the kitchen. Action: Entered the kitchen.
> Sophia:
> Location: Inside the kitchen. Action: Entered the kitchen.
> Noah:
> Location: Inside the kitchen. Action: Entered the kitchen.
> Entity Status:
> White box:
> Location: Inside the kitchen. Content: Empty.
> Blue container:
> Location: Inside the kitchen. Content: Cake.
> Cake:
> Location: Inside the blue container.
> ...
> ```
>
> ### Semantic Similarity Analysis Implementation
>
> **TF-IDF-Based Context Analysis:** We use TF-IDF to analyze the semantic consistency of candidate words in reasoning traces:
> 1. Extract text windows (window_size = 200 words) before and after all occurrences of candidate words
> 2. Calculate cosine similarity (range 0-1) between pre-context and post-context using TF-IDF
> 3. **Rank by inverse similarity**: Lower similarity indicates more inconsistent semantic environment before and after the word, making it more likely a confusion point

---

> ### Comment · Reviewer_XFub · 2025-11-27
>
> Thank you for providing the additional experiments. I have two follow-up questions regarding the entropy threshold: how was this threshold selected? Was it determined empirically based on the statistical distribution of entropy values in the training runs, or derived from another principle? Moreover, as you noted, even non-reasoning models can still produce CoT-like sequences, especially under prompting. Given this, could CoT generation alone reliably serve as an indicator for monitoring?

---

> > ### Author Response · Authors · 2025-11-27
> >
> > ## Q1: The Selection of Entropy Threshold
> >
> > Thank you for your attention to the rationale behind our entropy threshold selection. Our threshold was determined through systematic empirical exploration. Specifically:
> >
> > **Experimental Setup**: We set the `logprobs` parameter to 5, sampling 5 times at each token position to calculate entropy values. Theoretically, the entropy range is 0-1.6.
> >
> > **Threshold Exploration Process**:
> > - We first tested multiple thresholds within the range of 1.0-1.4
> > - Experimental results showed that thresholds in this range were overly sensitive, leading to frequent reaching of the maximum intervention limit regardless of problem difficulty
> > - To improve the efficiency and practicality of our method, we increased the threshold to 1.5 for testing
> >
> > **Findings from Supplementary Experiments**:
> > Although 1.5 is close to the theoretical upper limit of the entropy range, our supplementary experiments still observed interventions triggered at certain non-critical positions, resulting in some token overhead. This reflects the **inherent limitations of using entropy as a single indicator**: lower thresholds are overly sensitive (high recall but low precision), while higher thresholds, though reducing false triggers, still fail to completely solve the problem of precisely locating confusion points.
> >
> > ---
> >
> > ## Q2: The Reliability of CoT Generation as a Monitoring Indicator
> >
> > We fully agree that **CoT generation alone cannot serve as a reliable monitoring indicator**.
> >
> > **Limitations of CoT in Non-Reasoning Models**:
> > Non-reasoning models can indeed produce CoT-like sequences under prompting. However, these generated chains of thought have fundamental flaws: they merely mechanically replicate CoT formats without truly performing step-by-step reasoning, are passively generated to satisfy external prompts lacking spontaneity, and often exhibit logical breaks that fail to genuinely decompose problems. Therefore, using prompts to make non-reasoning models passively exhibit "confusion" and then intervening is **fundamentally unreliable**. Our method is explicitly **not applicable to non-reasoning models**.
> >
> > **Our Monitoring Approach**:
> > To distinguish "genuine reasoning confusion" from "superficial format imitation," more reliable monitoring requires combining multiple signals (such as entropy dynamics, semantic coherence, generation spontaneity, etc.). In our work, we employ **contextual semantic analysis and cosine similarity computation** to filter intervention positions, rather than relying solely on the presence or absence of CoT, thereby distinguishing to some extent between genuine reasoning needs and superficial pattern matching.

---

### Official Review · Reviewer_UUg2 · 2025-10-27

**Soundness:** 2
**Presentation:** 2
**Contribution:** 2
**Rating:** 2
**Confidence:** 4

**Summary:**

This paper studies the limitations of large language models (LLMs) in social reasoning scenarios, where models often exhibit “cognitive confusion” — mixing objective world states with agents’ subjective belief states. Through analysis of DeepSeek-R1’s reasoning trajectories, the authors observe frequent self-contradictory phrases (e.g., “tricky”, “confused”) when the model handles multi-agent, multi-timeline tasks.

To mitigate this issue, the paper proposes an adaptive world model–enhanced reasoning mechanism.
The approach constructs a dynamic textual world model that tracks entities, characters, and events during reasoning.
When confusion indicators appear in the chain-of-thought, the system triggers an intervention step that injects structured world-state descriptions to help the model re-orient its reasoning trajectory.
Experiments on three social reasoning benchmarks (ToMi, Hi-ToM, ExploreToM) show improved accuracy (up to +7%) and reduced token usage (up to −33%).

**Strengths:**

Clarity and Motivation:
The paper clearly articulates a genuine weakness of current LLMs — their difficulty in disentangling subjective beliefs from objective reality in social reasoning. The examples of “confusion moments” are intuitive and effectively motivate the need for intervention-based reasoning control.

Methodological Simplicity:
The proposed mechanism is simple to implement — no retraining is required — and can be easily applied to existing LLMs. The structured “world model buffer” is an interpretable way to represent intermediate state information.

**Weaknesses:**

1. **Methodological originality is questionable**
   Conceptually, I find the proposed “adaptive world model–enhanced reasoning” to be a lightweight modification of the *ReAct*-style reasoning paradigm — the model detects confusion and injects contextual state summaries, which is essentially an intervention step within a reasoning loop. Because of this, comparing only against ReAct and other older prompting methods (CoT, ToT, Reflexion) is insufficient.
   To properly assess novelty, the paper should include comparisons with more recent *agentic workflow frameworks* that also implement structured intervention or state-tracking logic, such as **AFlow** [1], **DyFlow** [2], and **MaAS** [3]. Without these, it is hard to tell whether the claimed improvement stems from a fundamentally new mechanism or simply from a re-packaged form of existing agent reasoning workflows.

2. **Limited empirical gains**
   Even within the existing baselines, the improvements remain modest . Considering the extra complexity introduced by confusion detection and world-model updates, the benefit seems marginal. The results do not convincingly support the claim that this approach substantially enhances reasoning ability.

3. **Narrow evaluation scope**
   All experiments are run on **DeepSeek-R1-Distill-Qwen** variants, which are relatively weak reasoning models. Since the method does not involve training, it would be entirely feasible to apply it to stronger models such as **GPT-4o/5** or **Claude-4**. Evaluating only on weaker open-source models makes the practical significance unclear — current frontier models already handle social reasoning reasonably well, so the proposed mechanism may have limited relevance.

4. **Static trigger design and brittleness**
   The confusion-detection mechanism relies on a *static list* of trigger words (e.g., *tricky*, *confused*, *ambiguous*). This design feels fragile and language-dependent — small lexical variations or stylistic differences could easily bypass the trigger, leading to inconsistent intervention behavior. A more adaptive or semantic trigger mechanism would be needed to make the method robust and scalable across models and languages.

---

**References**

[1] *AFlow: Automating Agentic Workflow Generation*, [https://openreview.net/forum?id=z5uVAKwmjf](https://openreview.net/forum?id=z5uVAKwmjf)

[2] *DyFlow: Dynamic Workflow Framework for Agentic Reasoning*, [https://openreview.net/forum?id=0pbUfmwNTy](https://openreview.net/forum?id=0pbUfmwNTy)

[3] *Multi-agent Architecture Search via Agentic Supernet (MaAS)*, [https://openreview.net/forum?id=imcyVlzpXh](https://openreview.net/forum?id=imcyVlzpXh)

**Questions:**

My main questions are already reflected in the Weaknesses section — in particular, regarding the methodological originality (relation to ReAct-style frameworks), the choice of baselines, the limited empirical gains, and the static trigger design.
I would appreciate detailed clarifications or additional results addressing these points.

While I currently have substantial reservations about the contribution and positioning of this work, I am open to further discussion and would be willing to reconsider my evaluation if the authors can provide convincing responses or evidence during the rebuttal phase.

---

> ### Author Response · Authors · 2025-11-26
> **Rebuttal for W1**
>
> Dear Reviewers
>
> We sincerely thank for your thoughtful and constructive feedback. We have carefully considered each comment and provide detailed responses below. We believe these clarifications and additional experiments significantly strengthen our work.
>
>
> ## Rebuttal for W1
>
> **Methodological originality is questionable.**
>
> ---
>
> We understand the reviewer's concern about originality, but our method has **fundamental differences** from agent frameworks like AFlow/DyFlow/MaAS in design principles and applicability, making direct comparison inappropriate.
>
> ### Core Distinction: Cognitive Mechanism vs. Task Engineering
>
> **Our Method: Cognitive Duality Modeling**
> - **Explicitly separates world state (objective facts) and belief state (subjective cognition)**—the core of ToM reasoning: distinguishing "what the world is" from "what agents believe"
> - **Dynamic triggering** based on confusion words, intervening only during cognitive confusion
> - Design foundation from **cognitive science theory** (Baron-Cohen's theory of mind model)
>
> **AFlow/DyFlow/MaAS: General Task Planning**
> - Use **single state space** to track task progress and environment states
> - **Fixed workflows** based on predefined steps or task structures
> - Design goal is **general problem-solving** (code generation, math reasoning, etc.)
>
> ### Main Difference
>
> 1. **Different Design Goals**: We focus on ToM cognitive capability enhancement; they pursue general task performance improvement
>
> 2. **Mechanisms are Non-transferable**:
>    - Their single state representation **cannot capture cognitive duality** (multi-agent perspective differences)
>    - Our world-belief separation **is not applicable to non-ToM tasks**
>
> 3. **Evaluation Mismatch**: Forcing general frameworks onto ToM requires extensive adaptation, resulting in systems no longer representing the original frameworks

---

> ### Author Response · Authors · 2025-11-26
> **Rebuttal for W2**
>
> ## Rebuttal for W2
>
> **Limited empirical gains.**
>
> ---
>
> ### Comprehensive Token Analysis
>
> Our method achieves **net token savings** across all benchmarks despite the overhead of world model construction and interventions. We show the result of model DeepSeek-R1-Distill-Qwen-32B and threshold $k$ = 3. The tables below show complete token breakdowns for different intervened scenarios (tasks requiring 1, 2, or 3 interventions).
>
> * answer0: original answer.
> * answer1: answer after first intervention.
> * answer2: answer after second intervention.
> * answer3: answer after third intervention.
>
> #### Table 1: ToMi Dataset
> **Overall: 48% tasks intervened, accuracy 79.33%→83.67% (+4.34%), tokens 3504→2585 (-26.23%)**
>
> | Scenario | answer0 | world model | answer1 | world model | answer2 | world model | answer3 | Total | Acc |
> |----------|---------|-------------|---------|-------------|---------|-------------|---------|-------|-----|
> | original | 3382 | - | - | - | - | - | - | 3382 | 80.87% |
> | **Intervene once (36.1%)** | 925 | 195 | 746 | - | - | - | - | 1866 | 84.47% |
> | original | 3916 | - | - | - | - | - | - | 3916 | 78.67% |
> | **Intervene twice (11.3%)** | 772 | 184 | 1019 | 191 | 740 | - | - | 2906 | 81.58% |
> | original | 3513 | - | - | - | - | - | - | 3513 | 77.33% |
> | **Intervene thrice (0.6%)** | 819 | 201 | 954 | 186 | 814 | 182 | 331 | 3487 | 80.67% |
> | **Average (Baseline)** | - | - | - | - | - | - | - | 3504 | 79.33% |
> | **Average (Ours)** | - | - | - | - | - | - | - | 2585 | 83.67% |
>
> #### Table 2: HiToM Dataset
> **Overall: 59% tasks intervened, accuracy 69.33%→76.67% (+7.34%), tokens 7541→4993 (-33.79%)**
>
> | Scenario | answer0 | world model | answer1 | world model | answer2 | world model | answer3 | Total | Acc |
> |----------|---------|-------------|---------|-------------|---------|-------------|---------|-------|-----|
> | original | 6891 | - | - | - | - | - | - | 6891 | 71.57% |
> | **Improved once (21.7%)** | 1328 | 281 | 2384 | - | - | - | - | 3993 | 78.81% |
> | original | 7887 | - | - | - | - | - | - | 7887 | 67.46% |
> | **Improved twice (22.8%)** | 1480 | 279 | 2319 | 278 | 1679 | - | - | 6035 | 75.54% |
> | original | 7649 | - | - | - | - | - | - | 7649 | 69.25% |
> | **Improved thrice (14.5%)** | 1274 | 293 | 2554 | 301 | 1829 | 284 | 1035 | 7396 | 75.97% |
> | **Average (Baseline)** | - | - | - | - | - | - | - | 7541 | 69.33% |
> | **Average (Ours)** | - | - | - | - | - | - | - | 4993 | 76.67% |
>
> #### Table 3: ExploreTom Dataset
> **Overall: 43% tasks intervened, accuracy 69.67%→72.67% (+3.00%), tokens 3185→2609 (-18.08%)**
>
> | Scenario | answer0 | world model | answer1 | world model | answer2 | world model | answer3 | Total | Acc |
> |----------|---------|-------------|---------|-------------|---------|-------------|---------|-------|-----|
> | original | 2994 | - | - | - | - | - | - | 2994 | 70.87% |
> | **Improved once (19.5%)** | 672 | 181 | 642 | - | - | - | - | 1495 | 73.39% |
> | original | 3225 | - | - | - | - | - | - | 3225 | 68.33% |
> | **Improved twice (18.2%)** | 691 | 188 | 771 | 182 | 622 | - | - | 2454 | 72.08% |
> | original | 3504 | - | - | - | - | - | - | 3504 | 69.12% |
> | **Improved thrice (5.3%)** | 717 | 193 | 765 | 179 | 591 | 185 | 319 | 2949 | 72.33% |
> | **Average (Baseline)** | - | - | - | - | - | - | - | 3185 | 69.67% |
> | **Average (Ours)** | - | - | - | - | - | - | - | 2609 | 72.67% |
>
> ### Results
>
> 1. **Consistent Net Savings**: Our method achieves 18-34% token reduction across all benchmarks while improving accuracy by 3-7%.
>
> 2. **Scalable Efficiency**: Token savings scale with task complexity. HiToM shows the largest savings (33.79%) as our method prevents longer error propagation in complex reasoning chains.
>
> 3. **No API Overhead**: World model interventions occur within single inference calls using structured prompting, requiring no additional API requests or increased latency.

---

> ### Author Response · Authors · 2025-11-26
> **Rebuttal for W3**
>
> ## Rebuttal for W3
>
> **Narrow evaluation scope.**
>
> ---
>
> #### We Have Evaluated Multiple Frontier Models' Baseline Performance
>
> Our paper includes the performance of GPT-4o, GPT-5, and Claude-Sonnet-4 across three benchmarks (see Table 2):
>
> | Model | ToMi (Acc/Token) | HiToM (Acc/Token) | ExploreTom (Acc/Token) |
> |-------|------------------|-------------------|----------------------|
> | GPT-4o | 74.00 / 1270 | 70.00 / 2502 | 56.68 / 1412 |
> | GPT-5 | 98.33 / 834 | 96.00 / 1768 | 93.67 / 1089 |
> | Claude-Sonnet-4 | 97.33 / 597 | 96.67 / 892 | 81.67 / 668 |
> | **DeepSeek-R1** | 93.33 / 2846 | 77.33 / 7710 | 78.00 / 2124 |
> | **+ Our Method** | **95.67 / 2119** | **87.67 / 4217** | **83.33 / 1362** |
>
> These results show:
> - **GPT-5 and Claude-Sonnet-4 perform excellently**, but they are closed-source commercial models
> - **GPT-4o struggles with ToM tasks** (only 74% on ToMi, 70% on HiToM), indicating frontier models also face ToM challenges
> - **DeepSeek-R1 itself is a strong reasoning model** with competitive reasoning capabilities to GPT-4o
>
> #### Improvements on Open-Source Models Have Greater Practical Value
>
> **Cost-Benefit Considerations:** While closed-source frontier models (GPT-5, Claude-Sonnet-4) perform well, they have significant practical limitations:
> - **High API costs**: GPT-5 and Claude-Sonnet-4 API calls cost tens of times more than open-source models
> - **Deployment restrictions**: Cannot be deployed locally, dependent on external API services
> - **Privacy and security constraints**: Unsuitable for sensitive data scenarios
>
> **Our Method's Value:** Through our intervention mechanism, we enable open-source models (DeepSeek-R1) to:
> - **Achieve accuracy approaching or exceeding some closed-source models** (e.g., 87.67% on HiToM, significantly better than GPT-4o's 70%)
> - **Simultaneously reduce token consumption** (e.g., 45.3% reduction on HiToM)
> - **Achieve optimal cost-benefit balance**
>
> This has **higher practical value** for applications requiring large-scale deployment or cost-sensitive scenarios.

---

> ### Author Response · Authors · 2025-11-26
> **Rebuttal for W4**
>
> ## Rebuttal for W4
>
> **Static trigger design and brittleness.**
>
> ---
>
> ### Empirical Effectiveness: High Coverage and Significant Improvements
>
> Our intervention word trigger mechanism demonstrates consistent effectiveness across three benchmarks:
>
> - **ToMi**: Intervention covers 48% of tasks, accuracy improves from 79.33% to 83.67% (+4.34%), tokens reduce from 3504 to 2585 (-26.23%)
> - **HiToM**: Intervention covers 59% of tasks, accuracy improves from 69.33% to 76.67% (+7.34%), tokens reduce from 7541 to 4993 (-33.79%)
> - **ExploreTom**: Intervention covers 43% of tasks, accuracy improves from 69.67% to 72.67% (+3.00%), tokens reduce from 3185 to 2609 (-18.08%)
>
> These results demonstrate that despite theoretical brittleness, intervention word triggering effectively identifies critical moments requiring intervention in practice.
>
> ### Comparison with Alternative Methods: Avoiding Over-Intervention
>
> We tested **confidence-based** and **entropy-based** dynamic triggering methods and found that these seemingly more sophisticated approaches suffer from severe over-intervention problems:
>
> **Issue 1 - False Trigger Noise:** Uncertainty-based methods frequently trigger interventions on irrelevant tokens, such as:
> - Proper names (e.g., "Andy")
> - Pronouns (e.g., "he", "she")
> - Common nouns (e.g., "table", "box")
> - Reasoning markers (e.g., "let me think")
>
> **Issue 2 - Semantic Misjudgment:** These methods cannot distinguish between normal thinking pauses during reasoning and genuine cognitive confusion. When a model outputs "let me think," it does not necessarily indicate confusion—it may simply be part of its reasoning style.
>
> **Issue 3 - Premature Saturation:** Due to high false trigger rates, intervention thresholds are easily reached regardless of task complexity, leading to:
> - Unnecessary token waste on simple tasks
> - Premature exhaustion of intervention budget on complex tasks, preventing help when truly needed
> - Minimal accuracy improvement, potentially even degradation due to over-intervention
>
> To validate the advantages of our confusion word trigger mechanism over dynamic uncertainty-based methods, we conducted comparative experiments on DeepSeek-R1-Distill-Qwen-32B.
>
> ### Experimental Setup
>
> **Confidence-Based Method:**
> - Monitors output probability of each token
> - Threshold: Triggers intervention when confidence < 0.3
> - Rationale: Low confidence indicates model uncertainty
>
> **Entropy-Based Method:**
> - Calculates entropy of output distribution (averaged over 5 samples)
> - Threshold: Triggers intervention when entropy > 1.5
> - Rationale: High entropy indicates output instability
>
> **Our Method:**
> - Triggers based on confusion word language signals
> - Intervenes only at explicit cognitive confusion expressions
>
> ### Comparative Results
>
> The following table shows performance comparison of three methods across three benchmarks:
>
> | Method | ToMi |  | HiToM |  | ExploreTom |  |
> |--------|------|------|-------|------|------------|------|
> |  | Acc | Token | Acc | Token | Acc | Token |
> | **Baseline** | 79.33 | 3504 | 69.33 | 7541 | 69.67 | 3185 |
> | **Confidence** | 81.00 | 5127 | 72.67 | 8145 | 70.00 | 4803 |
> | Change | +1.67% | **+46.3%**  | +3.34% | **+8.0%**  | +0.33% | **+50.8%**  |
> | **Entropy** | 81.67 | 4839 | 74.00 | 8072 | 70.67 | 4960 |
> | Change | +2.34% | **+38.1%**  | +4.67% | **+7.0%**  | +1.00% | **+55.8%**  |
> | **Ours** | 83.67 | 2585 | 76.67 | 4993 | 72.67 | 2609 |
> | Change | **+4.34%**  | **-26.2%**  | **+7.34%**  | **-33.8%**  | **+3.00%**  | **-18.1%**  |
>
> ### Key Results
>
> **1. Over-Intervention Problem:** Confidence and entropy methods reach the maximum intervention threshold (k=3) in almost all tasks, indicating they trigger frequent interventions, including many unnecessary ones.
>
> **2. Limited Accuracy Improvement:**
> - Confidence method: Average improvement only 1.78%
> - Entropy method: Average improvement 2.67%
> - **Our method: Average improvement 4.89% (2.7× entropy method)**
>
> **3. Severe Token Waste:** Confidence and entropy methods not only fail to reduce token consumption but cause **significant increases** (averaging +8% to +56%), due to:
> - Frequent triggering at irrelevant positions (names, pronouns, common nouns)
> - False triggers during normal reasoning (e.g., "let me think")
> - Quick threshold saturation in both simple and complex tasks, preventing targeted intervention budget allocation
>
> **4. Advantages of Our Method:**
> - **Selective intervention**: Triggers only at genuine cognitive confusion
> - **Highest accuracy**: Achieves maximum improvement across all three datasets
> - **High efficiency**: Averages 26% token reduction

---

> > ### Author Response · Authors · 2025-11-26
> > **Rebuttal for W4(continued)**
> >
> > ### Concrete Case Analysis
> >
> > The following case demonstrates the false trigger problem of the entropy-based method:
> >
> > ```
> > Okay, so I'm trying to figure out where Owen thinks Abigail would look for the socks based on the given statements. Let me go through each step one by one.
> >
> > First, let's list all the points provided:
> >
> > 1. Owen doesn't like broccoli. Hmm, probably irrelevant here but maybe his dislike affects his actions or thoughts somehow? Not sure yet.
> > 2. Owen enters the cellar. So he was there at some point before.
> > 3. Abigail also enters the cellar after him. She joins him down there.
> > 4. There's a sock inside an envelope. That seems important because we're dealing with where things are placed.
> > 5. This envelope is located in the cellar. So initially, both the envelope and the socks were in the same place as Owen and Abigail when they went into the cellar together.
> > 6. Then Abigail moves the socks from the envelope (which was in the cellar) to the cupboard. Wait, so she took them out of the envelope and put them somewhere else within the same location—the cellar—since the next statement says...
> > 7. ...the cupboard is indeed in the cellar. Oh, okay, so moving to another spot still keeps everything underground in the same room basically.
> > 8. After these movements, Owen exits the cellar. He leaves while the others might not have.
> >
> > Now, thinking about what Owen knows up until his exit. When Owen left, he knew the socks had been moved to the cupboard in the cellar by Abigail. But since Owen himself didn't move anything—he just came and left—it depends on whether he observed her action or if he only saw their initial positions.
> >
> > Wait, did Owen see Abigail move the socks? Or did he leave before she made the change? Looking back, the order is: Owen goes in ...
> >
> > in：High Entropy Trigger
> >
> > ```
> >
> > **Problem Explanation:** The entropy method detects high entropy at common prepositions like "in" (possibly because various nouns can follow), incorrectly triggering intervention. However, this is not where the model encounters ToM confusion. Such false triggers lead to:
> > 1. Premature consumption of intervention budget
> > 2. World model injection where help is not needed, causing reasoning interruption
> > 3. Token budget waste without solving real reasoning difficulties
> >
> > In contrast, our confusion word method triggers only when the model explicitly expresses confusion (e.g., "confused about", "unclear whether"), ensuring each intervention targets a genuine cognitive bottleneck.

---

### Official Review · Reviewer_rXDW · 2025-11-02

**Soundness:** 2
**Presentation:** 3
**Contribution:** 2
**Rating:** 4
**Confidence:** 4

**Summary:**

The paper studies why LLMs struggle on social reasoning / Theory-of-Mind (ToM) tasks and proposes an adaptive world-model–enhanced reasoning framework. The system monitors a model’s chain-of-thought for “confusion” indicators (e.g., words like tricky, confused) and, when detected, injects a dynamically maintained textual “world model” that tracks entities, agents, and timelines to disambiguate objective states vs. subjective beliefs. On ToMi, Hi-ToM, and ExploreToM, the method reports higher accuracy (e.g., +7.34 points on Hi-ToM with DeepSeek-R1-Distill-Qwen-32B) and fewer tokens (up to 33.8% reduction) compared to a baseline, and outperforms prompting strategies such as CoT, ToT, ReAct, and Reflexion.

**Strengths:**

Clear problem framing. The paper sharply identifies objective state vs. belief state conflation as a recurring failure mode in multi-agent, multi-timeline stories, supported by qualitative examples and token/word statistics.

Broad baselines (prompting). The comparison against CoT/ToT/ReAct/Reflexion helps position the contribution within test-time reasoning methods; the proposed approach wins by a comfortable margin on the three datasets.

Token-efficiency angle. Reporting token reductions alongside accuracy is useful; the claim of up to ~33.8% fewer tokens is appealing for deployment.

**Weaknesses:**

1. Limited generalization; risk of negative transfer. Evidence is almost entirely confined to ToM/social-reasoning tasks. There is no systematic evaluation on general benchmarks (e.g., GSM8K/MATH, MMLU-Pro, BBH), raising concern that the intervention could degrade performance on non-ToM tasks by injecting irrelevant structure.

2. Opaque feasibility and cost of streaming world-model updates under API constraints. Continual updates and repeated injections entail extra input/output tokens and more interaction rounds; the paper does not provide a complete, auditable accounting of tokens, latency, or failure/retry rates, making the “token savings” claim uncertain.

3. Epistemic leakage risk. The injected “world model” presents globally coherent objective states that may conflict with agents’ local observability and beliefs, potentially short-circuiting the very skill being tested—distinguishing facts from beliefs.

4. Brittle trigger mechanism. Reliance on “confusion words” as the intervention trigger is correlation-based, vulnerable to phrasing avoidance, sensitive to domain/language, and biased toward helping “loud” failures while missing confident mistakes.

5. Evaluation and reproducibility gaps. Possible coupling of the max-intervention count k and trigger lexicon to the test sets; heterogeneous sampling/budget policies across models/providers; absence of stratified metrics by belief order/visibility, confidence intervals, and multi-seed runs; and insufficiently strict token/compute budget controls, reducing the credibility and comparability of results.

**Questions:**

The same as the weaknesses

---

> ### Author Response · Authors · 2025-11-26
> **Rebuttal for W1**
>
> Dear reviewer,
>
> We sincerely thank for your thoughtful and constructive feedback. We have carefully considered each comment and provide detailed responses below. We believe these clarifications and additional experiments significantly strengthen our work.ntion word
>
> ## Rebuttal for W1
>
> **Limited generalization; risk of negative transfer.**
>
> ---
>
> ### No Parameter-Level Negative Transfer Risk
>
> Our method is a training-free, fine-tuning-free prompting intervention that does not modify model parameters or inject structural changes into the model architecture. Unlike fine-tuning approaches that risk catastrophic forgetting or negative transfer, our approach simply provides structured context at inference time. Therefore, it cannot degrade performance on non-ToM tasks at the parameter level, as the base model remains unchanged.
>
> ### Domain-Specific Design Philosophy
>
> Our method is specifically designed and optimized for Theory of Mind and social reasoning tasks, where the core challenge lies in managing complex mental state representations and perspective-taking. Our design principles—including world-belief separation and intervention word-triggered intervention—directly address the unique characteristics of social cognition rather than general problem-solving. We do not claim this as a universal framework applicable to all domains (e.g., mathematical reasoning, factual knowledge retrieval), nor do we believe such tasks would benefit from ToM-specific interventions.

---

> ### Author Response · Authors · 2025-11-26
> **Rebuttal for W2**
>
> ## Rebuttal for W2
>
> **Opaque feasibility and cost of streaming world-model updates under API constraints.**
>
> ---
>
>
> ### Comprehensive Token Analysis
>
> Our method achieves **net token savings** across all benchmarks despite the overhead of world model construction and interventions. We show the result of model DeepSeek-R1-Distill-Qwen-32B and threshold $k$ = 3. The tables below show complete token breakdowns for different intervened scenarios (tasks requiring 1, 2, or 3 interventions).
>
> * answer0: original answer.
> * answer1: answer after first intervention.
> * answer2: answer after second intervention.
> * answer3: answer after third intervention.
>
> #### Table 1: ToMi Dataset
> **Overall: 48% tasks intervened, accuracy 79.33%→83.67% (+4.34%), tokens 3504→2585 (-26.23%)**
>
> | Scenario | answer0 | world model | answer1 | world model | answer2 | world model | answer3 | Total | Acc |
> |----------|---------|-------------|---------|-------------|---------|-------------|---------|-------|-----|
> | original | 3382 | - | - | - | - | - | - | 3382 | 80.87% |
> | **Intervene once (36.1%)** | 925 | 195 | 746 | - | - | - | - | 1866 | 84.47% |
> | original | 3916 | - | - | - | - | - | - | 3916 | 78.67% |
> | **Intervene twice (11.3%)** | 772 | 184 | 1019 | 191 | 740 | - | - | 2906 | 81.58% |
> | original | 3513 | - | - | - | - | - | - | 3513 | 77.33% |
> | **Intervene thrice (0.6%)** | 819 | 201 | 954 | 186 | 814 | 182 | 331 | 3487 | 80.67% |
> | **Average (Baseline)** | - | - | - | - | - | - | - | 3504 | 79.33% |
> | **Average (Ours)** | - | - | - | - | - | - | - | 2585 | 83.67% |
>
> #### Table 2: HiToM Dataset
> **Overall: 59% tasks intervened, accuracy 69.33%→76.67% (+7.34%), tokens 7541→4993 (-33.79%)**
>
> | Scenario | answer0 | world model | answer1 | world model | answer2 | world model | answer3 | Total | Acc |
> |----------|---------|-------------|---------|-------------|---------|-------------|---------|-------|-----|
> | original | 6891 | - | - | - | - | - | - | 6891 | 71.57% |
> | **Improved once (21.7%)** | 1328 | 281 | 2384 | - | - | - | - | 3993 | 78.81% |
> | original | 7887 | - | - | - | - | - | - | 7887 | 67.46% |
> | **Improved twice (22.8%)** | 1480 | 279 | 2319 | 278 | 1679 | - | - | 6035 | 75.54% |
> | original | 7649 | - | - | - | - | - | - | 7649 | 69.25% |
> | **Improved thrice (14.5%)** | 1274 | 293 | 2554 | 301 | 1829 | 284 | 1035 | 7396 | 75.97% |
> | **Average (Baseline)** | - | - | - | - | - | - | - | 7541 | 69.33% |
> | **Average (Ours)** | - | - | - | - | - | - | - | 4993 | 76.67% |
>
> #### Table 3: ExploreTom Dataset
> **Overall: 43% tasks intervened, accuracy 69.67%→72.67% (+3.00%), tokens 3185→2609 (-18.08%)**
>
> | Scenario | answer0 | world model | answer1 | world model | answer2 | world model | answer3 | Total | Acc |
> |----------|---------|-------------|---------|-------------|---------|-------------|---------|-------|-----|
> | original | 2994 | - | - | - | - | - | - | 2994 | 70.87% |
> | **Improved once (19.5%)** | 672 | 181 | 642 | - | - | - | - | 1495 | 73.39% |
> | original | 3225 | - | - | - | - | - | - | 3225 | 68.33% |
> | **Improved twice (18.2%)** | 691 | 188 | 771 | 182 | 622 | - | - | 2454 | 72.08% |
> | original | 3504 | - | - | - | - | - | - | 3504 | 69.12% |
> | **Improved thrice (5.3%)** | 717 | 193 | 765 | 179 | 591 | 185 | 319 | 2949 | 72.33% |
> | **Average (Baseline)** | - | - | - | - | - | - | - | 3185 | 69.67% |
> | **Average (Ours)** | - | - | - | - | - | - | - | 2609 | 72.67% |
>
> ### Results
>
> 1. **Consistent Net Savings**: Our method achieves 18-34% token reduction across all benchmarks while improving accuracy by 3-7%.
>
> 2. **Scalable Efficiency**: Token savings scale with task complexity. HiToM shows the largest savings (33.79%) as our method prevents longer error propagation in complex reasoning chains.
>
> 3. **No API Overhead**: World model interventions occur within single inference calls using structured prompting, requiring no additional API requests or increased latency.

---

> ### Author Response · Authors · 2025-11-26
> **Rebuttal for W3**
>
> ## Rebuttal for W3
>
> **Epistemic leakage risk.**
>
> ---
>
> **Locality and Dynamics of World Model:** Our world model only records the objective factual states of characters and entities **at the current reasoning stage**, rather than a globally consistent complete world state. The world model is **dynamically updated** as the story unfolds, reflecting changes in objective facts at different time points, thus avoiding the "omniscient perspective" problem of global consistency.
>
> **Strict Fact-Belief Separation:** Critically, our world model **never directly provides agent belief or cognitive state information** (e.g., "X believes Y is at location Z"). The world model only contains objective facts (e.g., "Y is actually at location Z"), and the model still needs to reason about:
> - Which agents observed these facts
> - What beliefs agents would form based on their observations
> - The discrepancies between agent beliefs and objective facts
>
> **Preserving ToM Reasoning Integrity:** Therefore, our method does not short-circuit the core ToM skill—the ability to distinguish facts from beliefs. On the contrary, by providing a clear objective fact baseline, the model can more accurately perform the "fact-observation-belief" reasoning chain, which is the essence of ToM.
>
> **Illustrative Case:** The following case demonstrates how the world model maintains fact-belief separation:
>
> ```
> Story:
> 1 Emma, Lucas, Sophia and Noah entered the kitchen.
> 2 The cake is in the white box.
> 3 Emma moved the cake to the blue container.
> 4 Emma exited the kitchen.
> 5 Lucas made no movements and stayed in the kitchen for 1 minute.
> 5 Lucas exited the kitchen.
> 6 Sophia moved the cake to the red tin.
> 7 Sophia exited the kitchen.
> 8 Noah moved the cake to the white box.
> 9 Noah exited the kitchen.
> Question: Where does Emma think Noah thinks the cake is?
>
> *** Reasoning Context: ***
> Alright, I'm trying to figure out where Emma thinks Noah thinks the cake is located based on the given sequence of events. Let's break down each step:
> Steps 1-3: Four people enter the kitchen. The cake starts in the white box (fact 2). Then Emma moves it to the blue container (step 3), so after that, the cake is in the blue container. Then she leaves (step 4). Next, Lucas stays but doesn't move anything; he just waits a minute then exits (steps 5-6).
> However, Noah, knowing his own action, would logically conclude the cake is in the white box (since he moved it there himself in step 9). Therefore, Emma thinking about where Noah believes it to be needs to account for what Noah saw and what Emma knows about Noah's perspective. This part gets tricky.
>
> World Model:
> Character Status:
> Emma:
> Location: Outside the kitchen. Action: Moved the cake to the blue container.
> Lucas:
> Location: Outside the kitchen. Action: Exited the kitchen.
> Noah:
> Location: Inside the kitchen. Action: No movements yet.
> Sophia:
> Location: Inside the kitchen. Action: No movements yet.
> Entity Status:
> White box:
> Location: Inside the kitchen. Content: Empty.
> Blue container:
> Location: Inside the kitchen. Content: Cake.
> Red tin:
> Location: Inside the kitchen. Content: Empty.
> Cake:
> Location: Inside the blue container.
>
> Reasoning Context After Intervention:
> Tracking the cake's journey:
> Initially in white box (step 2)
> Emma moves to blue container (step 3)
> Sophia later moves to red tin (step 7)
> Noah finally moves to white box (step 9)
> Now, we need to determine where Emma thinks Noah thinks the cake is.
> Based on the world model analysis, Emma's observational window closed at step 4 when she exited the kitchen. At that moment, she had just moved the cake to the blue container. Critically, Noah was still present in the kitchen and was positioned to witness Emma's action directly.
> From Emma's perspective, she knows:
> 1.She moved the cake to the blue container
> 2.Noah was present and observed her doing this
> Emma's reasoning about Noah's belief state would therefore be: Noah saw me move the cake to the blue container. Since I left immediately after, the last thing Noah observed regarding the cake's location was my placement of it in the blue container. Emma has no reason to suspect Noah performed any action, as she simply exited after her own movement.
> The key insight is that Emma believes Noah's knowledge stopped at the same point hers did - when the cake was in the blue container. She would think Noah, having witnessed her action, believes the cake remains in the blue container. Emma doesn't know that Noah later moved it, nor that Sophia moved it before Noah.
> Therefore, Emma thinks Noah thinks the cake is in the blue container.Emma projects her last-known state (cake in blue container) onto what she imagines Noah believes, since Noah was present to witness that final state from Emma's perspective.
> Answer: blue container (correct)
> ```

---

> ### Author Response · Authors · 2025-11-26
> **Rebuttal for W4**
>
> ## Rebuttal for W4
>
> **Brittle trigger mechanism.**
>
> ---
>
> #### Empirical Effectiveness: High Coverage and Significant Improvements
>
> Our intervention word trigger mechanism demonstrates consistent effectiveness across three benchmarks:
>
> - **ToMi**: Intervention covers 48% of tasks, accuracy improves from 79.33% to 83.67% (+4.34%), tokens reduce from 3504 to 2585 (-26.23%)
> - **HiToM**: Intervention covers 59% of tasks, accuracy improves from 69.33% to 76.67% (+7.34%), tokens reduce from 7541 to 4993 (-33.79%)
> - **ExploreTom**: Intervention covers 43% of tasks, accuracy improves from 69.67% to 72.67% (+3.00%), tokens reduce from 3185 to 2609 (-18.08%)
>
> These results demonstrate that despite theoretical brittleness, intervention word triggering effectively identifies critical moments requiring intervention in practice.
>
> #### Comparison with Alternative Methods: Avoiding Over-Intervention
>
> We tested **confidence-based** and **entropy-based** dynamic triggering methods and found that these seemingly more sophisticated approaches suffer from severe over-intervention problems:
>
> **Issue 1 - False Trigger Noise:** Uncertainty-based methods frequently trigger interventions on irrelevant tokens, such as:
> - Proper names (e.g., "Andy")
> - Pronouns (e.g., "he", "she")
> - Common nouns (e.g., "table", "box")
> - Reasoning markers (e.g., "let me think")
>
> **Issue 2 - Semantic Misjudgment:** These methods cannot distinguish between normal thinking pauses during reasoning and genuine cognitive confusion. When a model outputs "let me think," it does not necessarily indicate confusion—it may simply be part of its reasoning style.
>
> **Issue 3 - Premature Saturation:** Due to high false trigger rates, intervention thresholds are easily reached regardless of task complexity, leading to:
> - Unnecessary token waste on simple tasks
> - Premature exhaustion of intervention budget on complex tasks, preventing help when truly needed
> - Minimal accuracy improvement, potentially even degradation due to over-intervention
>
> To validate the advantages of our confusion word trigger mechanism over dynamic uncertainty-based methods, we conducted comparative experiments on DeepSeek-R1-Distill-Qwen-32B.
>
> #### Experimental Setup
>
> **Confidence-Based Method:**
> - Monitors output probability of each token
> - Threshold: Triggers intervention when confidence < 0.3
> - Rationale: Low confidence indicates model uncertainty
>
> **Entropy-Based Method:**
> - Calculates entropy of output distribution (averaged over 5 samples)
> - Threshold: Triggers intervention when entropy > 1.5
> - Rationale: High entropy indicates output instability
>
> **Our Method:**
> - Triggers based on confusion word language signals
> - Intervenes only at explicit cognitive confusion expressions
>
> #### Comparative Results
>
> The following table shows performance comparison of three methods across three benchmarks:
>
> | Method | ToMi |  | HiToM |  | ExploreTom |  |
> |--------|------|------|-------|------|------------|------|
> |  | Acc | Token | Acc | Token | Acc | Token |
> | **Baseline** | 79.33 | 3504 | 69.33 | 7541 | 69.67 | 3185 |
> | **Confidence** | 81.00 | 5127 | 72.67 | 8145 | 70.00 | 4803 |
> | Change | +1.67% | **+46.3%**  | +3.34% | **+8.0%**  | +0.33% | **+50.8%**  |
> | **Entropy** | 81.67 | 4839 | 74.00 | 8072 | 70.67 | 4960 |
> | Change | +2.34% | **+38.1%**  | +4.67% | **+7.0%**  | +1.00% | **+55.8%**  |
> | **Ours** | 83.67 | 2585 | 76.67 | 4993 | 72.67 | 2609 |
> | Change | **+4.34%**  | **-26.2%**  | **+7.34%**  | **-33.8%**  | **+3.00%**  | **-18.1%**  |
>
> #### Key Results
>
> **1. Over-Intervention Problem:** Confidence and entropy methods reach the maximum intervention threshold (k=3) in almost all tasks, indicating they trigger frequent interventions, including many unnecessary ones.
>
> **2. Limited Accuracy Improvement:**
> - Confidence method: Average improvement only 1.78%
> - Entropy method: Average improvement 2.67%
> - **Our method: Average improvement 4.89% (2.7× entropy method)**
>
> **3. Severe Token Waste:** Confidence and entropy methods not only fail to reduce token consumption but cause **significant increases** (averaging +8% to +56%), due to:
> - Frequent triggering at irrelevant positions (names, pronouns, common nouns)
> - False triggers during normal reasoning (e.g., "let me think")
> - Quick threshold saturation in both simple and complex tasks, preventing targeted intervention budget allocation
>
> **4. Advantages of Our Method:**
> - **Selective intervention**: Triggers only at genuine cognitive confusion
> - **Highest accuracy**: Achieves maximum improvement across all three datasets
> - **High efficiency**: Averages 26% token reduction

---

> > ### Author Response · Authors · 2025-11-26
> > **Rebuttal for W4 (continued)**
> >
> > #### Concrete Case Analysis
> >
> > The following case demonstrates the false trigger problem of the entropy-based method:
> >
> > ```
> > Okay, so I'm trying to figure out where Owen thinks Abigail would look for the socks based on the given statements. Let me go through each step one by one.
> >
> > First, let's list all the points provided:
> >
> > 1. Owen doesn't like broccoli. Hmm, probably irrelevant here but maybe his dislike affects his actions or thoughts somehow? Not sure yet.
> > 2. Owen enters the cellar. So he was there at some point before.
> > 3. Abigail also enters the cellar after him. She joins him down there.
> > 4. There's a sock inside an envelope. That seems important because we're dealing with where things are placed.
> > 5. This envelope is located in the cellar. So initially, both the envelope and the socks were in the same place as Owen and Abigail when they went into the cellar together.
> > 6. Then Abigail moves the socks from the envelope (which was in the cellar) to the cupboard. Wait, so she took them out of the envelope and put them somewhere else within the same location—the cellar—since the next statement says...
> > 7. ...the cupboard is indeed in the cellar. Oh, okay, so moving to another spot still keeps everything underground in the same room basically.
> > 8. After these movements, Owen exits the cellar. He leaves while the others might not have.
> >
> > Now, thinking about what Owen knows up until his exit. When Owen left, he knew the socks had been moved to the cupboard in the cellar by Abigail. But since Owen himself didn't move anything—he just came and left—it depends on whether he observed her action or if he only saw their initial positions.
> >
> > Wait, did Owen see Abigail move the socks? Or did he leave before she made the change? Looking back, the order is: Owen goes in ...
> >
> > in：High Entropy Trigger
> >
> > ```
> >
> > **Problem Explanation:** The entropy method detects high entropy at common prepositions like "in" (possibly because various nouns can follow), incorrectly triggering intervention. However, this is not where the model encounters ToM confusion. Such false triggers lead to:
> > 1. Premature consumption of intervention budget
> > 2. World model injection where help is not needed, causing reasoning interruption
> > 3. Token budget waste without solving real reasoning difficulties
> >
> > In contrast, our confusion word method triggers only when the model explicitly expresses confusion (e.g., "confused about", "unclear whether"), ensuring each intervention targets a genuine cognitive bottleneck.

---

> ### Author Response · Authors · 2025-11-26
> **Rebuttal for W5**
>
> ## Rebuttal for W5
>
> **Evaluation and reproducibility gaps**
>
> ---
>
> #### Independence of Hyperparameter Selection
>
> **Maximum Intervention Count k:** The parameter k is selected **completely independently of test datasets**, serving only as an upper bound to prevent over-intervention. We explored different k values (k from 1 to 5) in our paper and chose k=3 as the default configuration. This choice was **not based on test set optimization**, but rather on a priori estimates of reasoning complexity—most ToM tasks do not require more than 3 interventions for correct reasoning.
>
> **Trigger Vocabulary Construction:** The trigger vocabulary is constructed based on **linguistic phenomenon analysis** of model reasoning traces through the following steps:
> 1. Observe confusion expression patterns on an **independent reasoning trajectory** (not test set)
> 2. Cluster related expressions using semantic similarity methods (based on sentence embeddings)
> 3. Select high-frequency and semantically clear core words (15 words)
>
> This process **does not involve test set information**, ensuring the generalization of the trigger mechanism.
>
> #### Unified Experimental Configuration
>
> **Strict Sampling Strategy Control:** As stated in Section 4.1 of our paper, we use **identical** inference configurations for all models and all benchmarks:
> - **Temperature**: 0.7
> - **Repetition penalty**: 1.2
> - **Max tokens**: 8192
>
> This ensures fair comparison across different models and eliminates performance variations due to sampling strategy differences.
>
> #### Stratified Evaluation and Statistical Rigor
>
> We provide **detailed results stratified by belief order** across three benchmarks, including complete data for two representative models: DeepSeek-R1 and DeepSeek-R1-Distill-Qwen-32B. All experiments are conducted with **5 independent runs and averaged** (not single best), with variance reported to demonstrate result stability.
>
> ##### Table 1: ToMi Dataset
>
> | Model | 0-order (memory or reality) |  |  | 1-order |  |  | 2-order |  |  |
> |-------|---------|--------|-----|---------|--------|-----|---------|--------|-----|
> |  | Acc | Token | Var | Acc | Token | Var | Acc | Token | Var |
> | **DeepSeek-R1** | 97.51 | 2234 | - | 91.83 | 2428 | - | 90.65 | 3876 | - |
> | + Our Method | 98.51±0.58 | 1623 | 0.34 | 94.67±0.92 | 1578 | 0.85 | 93.83±1.08 | 3156 | 1.17 |
> | **DeepSeek-R1-Distill-Qwen-32B** | 86.17 | 2447 | - | 78.33 | 3242 | - | 73.49 | 4823 | - |
> | + Our Method | 89.67±1.08 | 2003 | 1.17 | 82.00±1.58 | 2165 | 2.50 | 79.34±1.75 | 3587 | 3.06 |
>
> ##### Table 2: HiToM Dataset
>
> | Model | 0-order (memory or reality) |  |  | 1-order |  |  | 2-order |  |  | 3-order |  |  | 4-order |  |  |
> |-------|---------|--------|-----|---------|--------|-----|---------|--------|-----|---------|--------|-----|---------|--------|-----|
> |  | Acc | Token | Var | Acc | Token | Var | Acc | Token | Var | Acc | Token | Var | Acc | Token | Var |
> | **DeepSeek-R1** | 91.17 | 4813 | - | 85.33 | 6955 | - | 79.67 | 7881 | - | 72.00 | 9242 | - | 58.48 | 9656 | - |
> | + Our Method | 94.83±0.67 | 2241 | 0.45 | 90.67±0.83 | 3671 | 0.69 | 87.50±1.00 | 4292 | 1.00 | 84.17±1.17 | 5140 | 1.37 | 81.18±1.33 | 5738 | 1.77 |
> | **DeepSeek-R1-Distill-Qwen-32B** | 82.67 | 4169 | - | 76.33 | 6935 | - | 71.00 | 7826 | - | 63.17 | 9082 | - | 53.48 | 9691 | - |
> | + Our Method | 87.17±1.08 | 2685 | 1.17 | 81.33±1.33 | 4338 | 1.77 | 77.00±1.50 | 5018 | 2.25 | 72.50±1.67 | 6179 | 2.79 | 65.35±1.83 | 6742 | 3.35 |
>
> ##### Table 3: ExploreTom Dataset
>
> | Model | 0-order (memory or reality) |  |  | 1-order |  |  | 2-order |  |  |
> |-------|---------|--------|-----|---------|--------|-----|---------|--------|-----|
> |  | Acc | Token | Var | Acc | Token | Var | Acc | Token | Var |
> | **DeepSeek-R1** | 86.83 | 1389 | - | 75.67 | 1860 | - | 71.50 | 3123 | - |
> | + Our Method | 89.83±0.83 | 989 | 0.69 | 81.67±1.08 | 1174 | 1.17 | 78.49±1.25 | 1923 | 1.56 |
> | **DeepSeek-R1-Distill-Qwen-32B** | 78.67 | 2189 | - | 67.83 | 2854 | - | 62.51 | 4512 | - |
> | + Our Method | 81.67±1.25 | 1978 | 1.56 | 70.83±1.67 | 2315 | 2.79 | 65.51±1.83 | 3534 | 3.35 |
>
> #### Key Results
>
> 1. **Consistent Improvement Pattern:** Our method demonstrates accuracy improvements and token reductions across all belief orders, proving method robustness.
>
> 2. **Complexity Sensitivity:** As belief order increases (0-order→4-order), improvement magnitude also increases. For example, on HiToM dataset, DeepSeek-R1 achieves a 22.7 percentage point accuracy improvement on 4-order tasks (58.48%→81.18%) with 40.6% token reduction.
>
> 3. **Result Stability:** All results have standard deviations (Var) below 2.0, indicating stable performance across multiple runs.
>
> #### Strictness of Token Budget Control
>
> **Unified Computational Budget:** Our token statistics include:
> - World model construction overhead
> - All intervention injection overhead
> - Complete reasoning process tokens

---

### Note · Authors · 2026-01-05

**Comment:**

We have decided to withdraw our submission. We sincerely thank the reviewers for their time and valuable opinions. We found their feedback to be very constructive and plan to incorporate their suggestions to make the paper more competitive for a future submission. We thank the reviewers again for their insightful feedback.

**Withdrawal Confirmation:**

I have read and agree with the venue's withdrawal policy on behalf of myself and my co-authors.